# A comparison of 27 *Arabidopsis thaliana* genomes and the path toward an unbiased characterization of genetic polymorphism

Anna A. Igolkina[1,10], Sebastian Vorbrugg[2,10], Fernando A. Rabanal[2,10], Hai-Jun Liu [1,10], Haim Ashkenazy [2,10], Aleksandra E. Kornienko[1,10], Joffrey Fitz [2], Max Collenberg[2], Christian Kubica [2], Almudena Mollá Morales [1], Benjamin Jaegle[1], Travis Wrightsman[2], Vitaly Voloshin [3], Alexander D. Bezlepsky[4], Victor Llaca [5], Viktoria Nizhynska[1], Ilka Reichardt [1], Ilja Bezrukov[2], Christa Lanz[2], Felix Bemm[2], Pádraic J. Flood [6], Sileshi Nemomissa[7], Angela Hancock [6], Ya-Long Guo [8], Paul Kersey[3], Detlef Weigel [2,9] ✉ & Magnus Nordborg [1] ✉

Making sense of whole-genome polymorphism data is challenging, but it is essential for overcoming the biases in SNP data. Here we analyze 27 genomes of *Arabidopsis thaliana* to illustrate these issues. Genome size variation is mostly due to tandem repeat regions that are difficult to assemble. However, while the rest of the genome varies little in length, it is full of structural variants, mostly due to transposon insertions. Because of this, the pangenome coordinate system grows rapidly with sample size and ultimately becomes 70% larger than the size of any single genome, even for $n = 27$. Finally, we show how short-read data are biased by read mapping. SNP calling is biased by the choice of reference genome, and both transcriptome and methylome profiling results are affected by mapping reads to a reference genome rather than to the genome of the assayed individual.

The last 25 years have witnessed an explosion of genetic polymorphism data, fueled by Human Genome Project-inspired collaborations and the development of massively parallel technologies for sequencing and genotyping. Such data allow us to study population history, selection and genetic architecture of traits, as well as the evolution of the genome itself. However, our current view of genetic polymorphism has been shaped by technologies that attempt to align short sequence fragments to a reference genome to detect sites that differ. As a result, our knowledge of genome variation has remained incomplete and biased towards simple variants in regions that are easy to align—a small

fraction of the genome in many species. A further source of bias arises from the use of a single reference genome.

All this is beginning to change now that long reads are making it possible to assemble high-quality, full-length chromosomal sequences from population samples. During the last couple of years, nearly complete genomes have been produced for large numbers of eukaryotic species, including yeast, animals (including fruit flies, humans and cichlids) and many plants, including rice, tomato, soybean, grapevine, wheat, barley, maize, millet, *Brassica oleracea*, *Eucalyptus*, *Populus* and *Marchantia* sp.—as well as *Arabidopsis thaliana*[1–27]. Impressive as these

[1]Gregor Mendel Institute, Austrian Academy of Sciences, Vienna, Austria. [2]Max Planck Institute for Biology Tübingen, Tübingen, Germany. [3]Royal Botanic Gardens Kew, London, UK. [4]All-Russian Research Institute of Agricultural Microbiology, Saint Petersburg, Russia. [5]Corteva Agriscience, Johnston, IA, USA. [6]Max Planck Institute for Plant Breeding Research, Cologne, Germany. [7]Addis Ababa University, Addis Ababa, Ethiopia. [8]Institute of Botany, Chinese Academy of Sciences, Beijing, China. [9]Institute for Bioinformatics and Medical Informatics, University of Tübingen, Tübingen, Germany. [10]These authors contributed equally: Anna A. Igolkina, Sebastian Vorbrugg, Fernando A. Rabanal, Hai-Jun Liu, Haim Ashkenazy, Aleksandra E. Kornienko. ✉e-mail: weigel@tue.mpg.de; magnus.nordborg@gmi.oeaw.ac.at

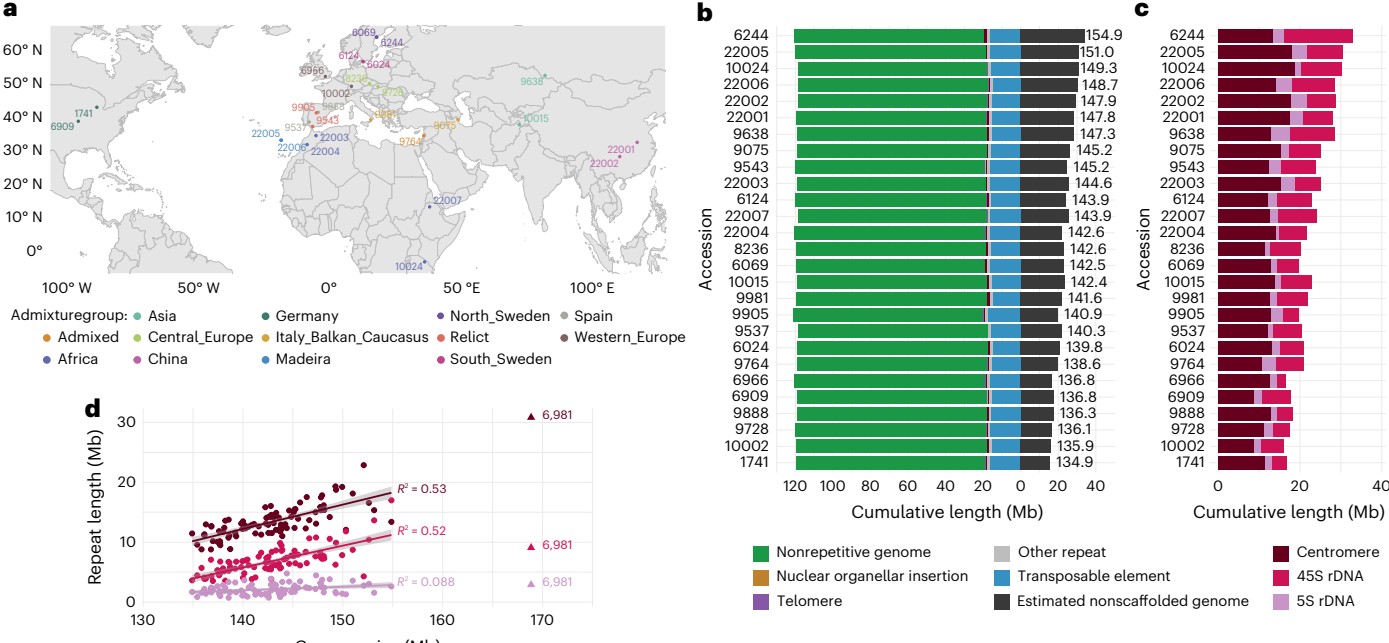

**Fig. 1 | Genome assemblies and size variation across 27 *A. thaliana* accessions.**
**a**, Origin of the sequenced accessions with colors indicating 'Admixture' group[34]. The reference accession Col-0 (6909) lacks reliable collection data (but hails from Central Europe based on genotype as well as historical records). Please note that 22005 and 22006 appear as a single point as they are geographically close and belong to the same admixture group. Maps were generated using public domain data from the Natural Earth project via the R package rnaturalearth. **b**, Histogram of genome sizes estimated from *k*-mers in PCR-free short reads with total assembly sizes superimposed (values to the left of 'zero'). Most of the variation in genome size can be attributed to unassembled regions (values to the right of 'zero'). **c**, The amount of centromeric, 5S rDNA and 45S rDNA repeats, estimated from PCR-free short reads with a BLAST-based approach. **d**, Correlation between genome size and each of the three major satellite repeats. Their estimated amounts jointly explain up to 92% ($P < 2.2 \times 10^{-16}$) of total genome size variation (for details, see Supplementary Note 3–'Estimation of satellite repeats'). The linear regression lines with shaded 95% confidence intervals exclude accession 6981 (Ws-2; indicated with a triangle) because of its very high centromeric repeat content.

studies are, they have also highlighted the difficulty in making sense of whole-genome polymorphism data, primarily because sequence alignment is not unambiguous. Pangenome graphs[28,29] may provide elegant and computationally efficient ways of representing such data, but they do not solve this fundamental problem. To compare genomes and interpret their differences properly, we require a modeling framework that reflects the mutational mechanisms and recombination history that gave rise to these differences, but such a framework is still largely missing.

Here we illustrate this problem by analyzing the genomes of 27 natural inbred accessions of *A. thaliana*, chosen to cover the global genetic diversity of the species. Our focus is on obtaining an unbiased picture of polymorphism in the more easily alignable chromosome arms and comparing it to existing data built over almost two decades[30–34]. To provide an unbiased picture of the 'gene-ome', that is, the collection of genes across multiple genomes, we complement our genome assemblies with transcriptomes from multiple tissues for the entire sample. In addition, we seek to lay the foundation for a community resource that will eventually comprise complete genomes for the thousands of natural inbred accessions that are publicly available for this model plant, thus connecting whole-genome polymorphism data to experimentally accessible germplasm and knowledge of a wide range of morphological, life-history, physiological and molecular phenotypes, as well as precise collection information (https://1001genomes.org/).

## Results

### The organization of genome variation

We selected 27 accessions to cover global genetic variation of the species based on the original 1001 Genomes Project[34], and additional samples from eastern Asia[35], Africa[36] and Madeira[37] (Fig. 1a). The genomes were sequenced using PacBio continuous long reads (CLRs) and

assembled as described in Methods and Supplementary Note 3–'The organization of genome variation').

Like many plants, *A. thaliana* has experienced recent episodes of transposable elements (TE) activity, leading to nearly identical sequences inserted across each genome[38]. These make short-read alignments difficult, but our CLR reads were long enough to bridge such insertions, allowing us to assemble the gene-dense chromosome arms with ease. However, extensive tracts of identical or near-identical tandem repeats, such as centromere satellites and rDNA, consistently break our assemblies (Supplementary Figs. 1 and 2). Of note, centromeres can now be assembled using PacBio HiFi reads[16], whereas 45S rDNA clusters remain challenging[39].

Our assemblies are all ~120 Mb in size, whereas the full genomes are estimated to range from 135 to 155 Mb (Fig. 1b), consistent with previous results[40,41]. This genome size variation appears to be almost entirely due to variation in centromeric and rDNA repeats (Fig. 1b–d), with the importance of 45S rDNA variation having been appreciated before[41]. While individual TE families can vary greatly in size across accessions (Supplementary Fig. 3), we confirm that the cumulative effect of all TEs on genome size variation appears to be small in *A. thaliana*[41]—contrary to the major role they have in interspecific variation[42,43].

### Detecting structural variation

In agreement with others[23], we find the chromosome arms were not only similar in length across accessions but also largely syntenic (Fig. 2a). Thirteen Mb-scale rearrangements were readily apparent in whole-genome alignments (WGAs). These include the previously described 1.2-Mb paracentric inversion associated with a heterochromatic knob on chromosome 4 (refs. 44,45; Extended Data Fig. 1 and Supplementary Table 2), and a very large reciprocal translocation in accession 22001 from the Yangtze River region (Supplementary Fig. 4).

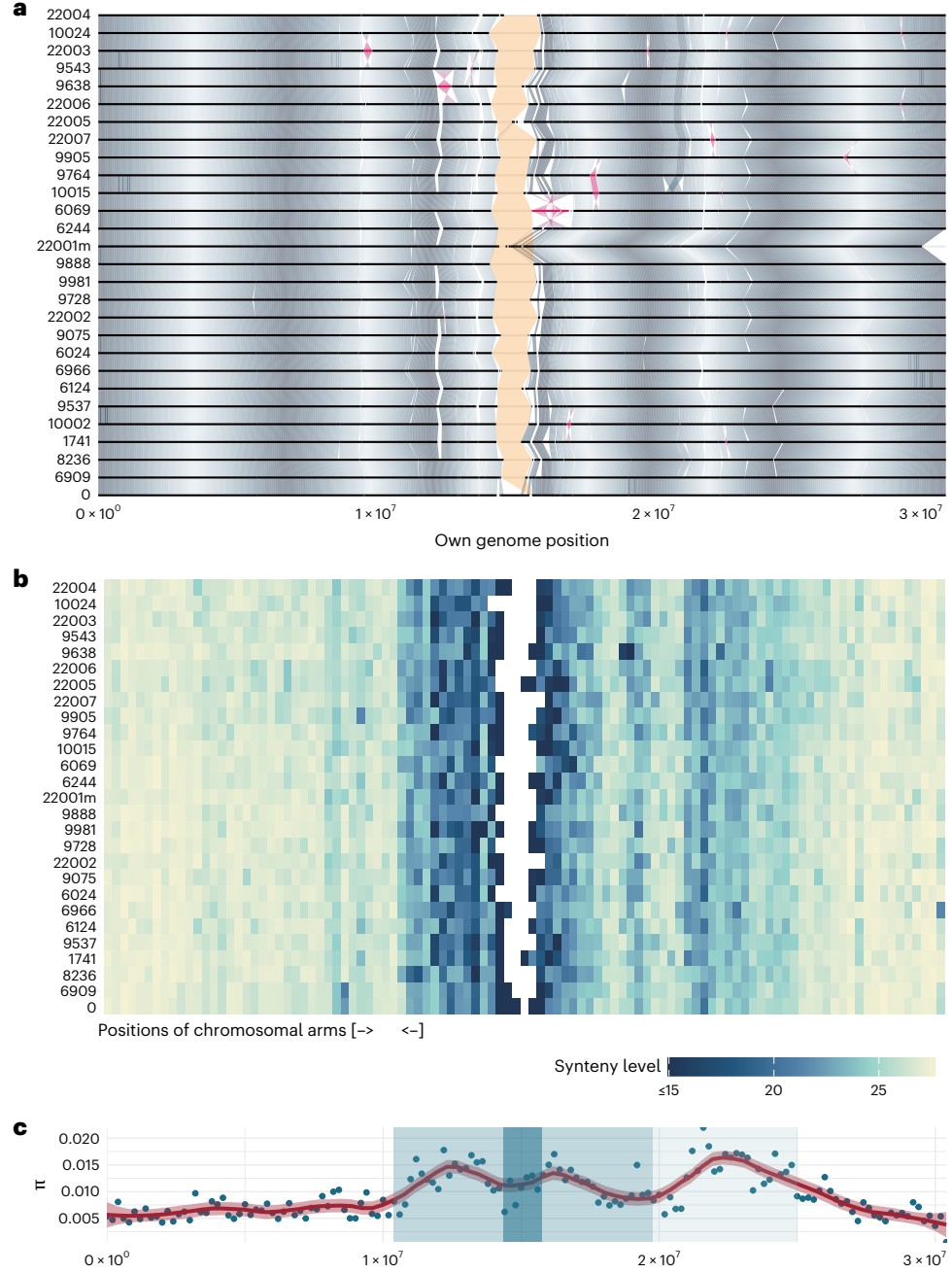

**Fig. 2 | Distribution of variation on chromosome 1. a**, WGA illustrates that the genomes are structurally conserved away from centromeric regions. Chromosomes are aligned from both ends to emphasize the contiguity of the arms, with the unassembled centromeric regions indicated in yellow and inversions in pink. Oscillating gray shades highlight homologous regions (the periodicity of the gradients reflects repeated use of the color scheme and has no biological meaning). **b**, The density of pangenome graph bubbles, reflecting higher diversity in pericentromeric regions and at ~20 Mb, where an ancestral centromere was lost through chromosome fusion[41]. Synteny level refers to the average sharing of links between nodes across the 27 genomes in 300-kb blocks. **c**, Distribution of nucleotide diversity based on SNPs called from a multiple alignment. Blue dots represent values of diversity in 200-kb windows. The red line is a smoothed fit to these points, with the standard error shown. The dark blue area corresponds to the centromeric region, with the lighter blue area highlighting the pericentromeric area and the lightest blue the ancestral lost centromere[41]. For chromosomes 2–5, see Extended Data Fig. 1.

For convenience, we treat this latter rearrangement as if it had not happened in the analysis below, and refer to the modified assembly as 22001m (Supplementary Note 3–'Reciprocal translocation in 22001').

Large-scale rearrangements aside, a comprehensive characterization of structural variants (SVs; by which we mean any alteration that causes variation in length, orientation or local context of sequence) remains difficult. The characterization of SVs is a fundamentally different problem from SNP calling. The latter can be viewed as a technical issue—namely, how to distinguish SNPs from sequencing errors—but

SV calling remains challenging even with flawless chromosomal sequences. The reason is that the SVs identified between genomes depend on the alignment method and parameters used, and there is no obvious ground truth. Therefore, we pursued two complementary approaches: a new whole-genome multiple-alignment pipeline, Pannagram[46], and the pangenome graph builder (PGGB)[29]. A discussion and comparison of these methods can be found in Supplementary Note 4. We emphasize that the differences between two algorithms designed to perform different tasks should not be interpreted as bias, and that,

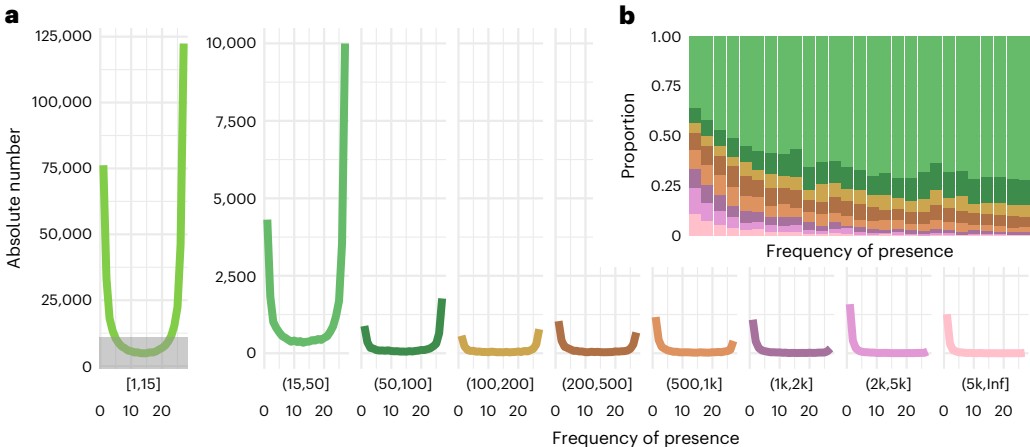

**Fig. 3 | The allele frequency distribution of SVs. a**, The frequency distribution of the presence alleles of all sSVs grouped by variant length (in bp). The height of the gray block (left) is equivalent to the height of the complete panel (right). **b**, The proportion of length classes of sSVs in each frequency bin. Colors correspond to those in **a**.

as noted above, there is no definitive ground truth. However, because Pannagram produces easily interpretable SVs along with convenient pangenome coordinates, and PGGB SVs cover all Pannagram SVs, we based our further SV analyses on the Pannagram data.

## Characterization of SVs

SVs come in many types and sizes, reflecting diverse mutational mechanisms. We will focus on length variants, which are by far the most numerous (for example, we identified fewer than a hundred inversions). We further divide them into bi-allelic presence/absence polymorphisms, consistent with simple insertion/deletion (indel) mutations (sSV), and more complex multi-allelic polymorphisms (cSV). We primarily consider the former, as they are easier to interpret and constitute the majority (over 80% overall), especially in the chromosome arms (Extended Data Fig. 2).

We identified 532,178 sSVs, affecting over 37.5 Mb in total—with the length distribution being heavily skewed towards short variants (Fig. 3). To gain further insight into the nature of these polymorphisms, we consider sSVs between 15 and 20,000 bp length, because shorter variants are more sensitive to the choice of alignment parameters, and larger ones are too few for statistical treatment. The frequency distribution for the presence allele of sSVs was consistent with sSVs mostly being due to rare insertions or rare deletions. Specifically, long presence alleles tend to be rare, and short ones common, suggesting that sSVs are mainly caused by long insertions and short deletions. Both types are also more likely to occur in intergenic regions than in genic regions, and they are more often observed in introns than in exons, consistent with purifying selection removing many of them (Extended Data Fig. 2).

Of particular note are 108 organellar insertions, almost all of them singletons or doubletons, ranging from a few hundred bp to entire organellar genomes (Supplementary Fig. 7).

## SVs and annotated TEs

Active TEs produce SVs of diverse natures, including both insertions and deletions, and serve as templates for various mutational processes, such as double-stranded break repair[47]. We classified the presence alleles of 17,447 sSVs of length ≥100 bp based on their BLAST-identified coverage by ~35,000 annotated TE sequences, spanning ~15% of the *A. thaliana* reference genome (Fig. 1b). In total, over 60% of sSVs showed some overlap with TE annotation, confirming a strong connection between our sSVs and TEs (Fig. 4b).

Likely insertions (presence allele in 1–3 accessions) tend to be longer than likely deletions (absence allele in 1–3 accessions). As shown in Fig. 4c,d, likely insertions also correspond more often to complete TEs, consistent with recent TE activity, whereas likely deletions more often correspond to incomplete TEs, suggesting that they are decaying elements.

As expected, sSVs corresponding to complete TEs are enriched for particular lengths (specifically around 5 kb), reflecting activity of complete TEs (Fig. 4e). Similar patterns of enrichment were also found for sSVs in all other categories except for those corresponding to TE fragments. This supports recent reports that active TEs are far from perfectly annotated[48,49].

## The mobile-ome

Mobile elements that have been active in the history of our sample of genomes will have generated segregating insertions and deletions with similar sequences at different locations in the genomes. We can use this property to look directly for mobile elements without relying on TE annotation. We will refer to the set of such elements as the mobile-ome, noting that our usage differs somewhat from that of others[38]. To identify the mobile-ome, we used Pannagram to cluster all presence alleles from sSVs based on sequence similarity and represented the output as a graph of nestedness, where nodes represent sequences, and connected components are expected to correspond to distinct TE superfamilies or families. We note that several similar approaches have recently been proposed[50–52].

Almost all sSVs that corresponded to complete TEs are included in the graph, consistent with their being part of the mobile-ome (Fig. 5a). Among other TE-related sSVs, approximately 70% are linked to the graph and are thus also related to the mobile-ome, presumably reflecting a mixture of incompletely annotated TE families, complex insertion behaviors and deletions within active TE families.

To understand the relationship between annotated TEs, we filtered out sequences without TE content and merged sequences with high similarity into larger nodes. The resulting subgraph consists of one dominant component along with numerous smaller ones (Fig. 5b). As expected, there are multiple large nodes corresponding to complete TEs, but we also see large nodes corresponding to sequences ostensibly containing complete TEs or TE fragments, demonstrating that many of these sequences are in fact part of the mobile-ome (Extended Data Fig. 3). Coloring the nodes by TE superfamily reveals that, while the smaller components mostly can be attributed to single TE families, the dominant component contains members of all known TE superfamilies (Fig. 5c). Possible reasons include TEs inserting into existing TEs and closely related sequences being mis-annotated as belonging to

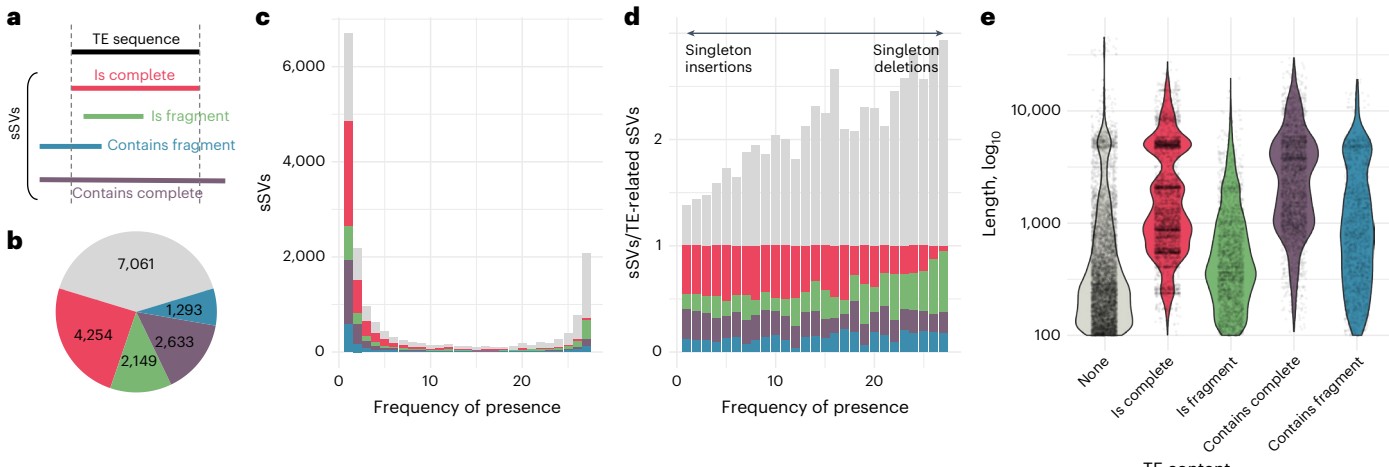

**Fig. 4 | The role of TEs in bi-allelic indels. a**, Categories of overlap between the presence allele of sSVs and annotated TE sequences. **b**, Number of sSVs in each TE-content category described in **a**. Colors correspond to **a**–gray, no TE content. **c**,**d**, Histograms of presence frequency showing annotated TE content as a function of frequency, either raw counts (**c**) or relative to TE-related SVs (**d**). **e**, Distribution of the length of presence alleles in different categories. For an analysis broken down by TE superfamily (Supplementary Fig. 12).

different superfamilies (for example, Supplementary Fig. 13). Further evidence for the incompleteness of the TE annotation comes from small graph components that contain large nodes of longer elements that have only partial TE content (Supplementary Figs. 14 and 15).

In the set of sSVs without TE content, the majority (74%) are unique and hence not connected in the graph. Among the remaining 26%, we observed 238 connected components with ≥3 nodes, similar in structure to those corresponding to single TE families, and thus presumably corresponding to previously unannotated TEs (Fig. 5d). Of these, 97 are obviously TE-like, encoding TE proteins from *A. thaliana* or other species. A few of them show evidence of horizontal gene transfer, as they exhibit greater protein similarity to species outside a panel of five Brassicaceae species (for example, Supplementary Fig. 16). However, we also found putative new mobile element families that lack clear protein-coding (PC) potential. They form relatively large components in the graph (Fig. 5d, purple islands), indicating that they are not rare. They are not low complexity, and some are exclusive to *A. thaliana* (see Supplementary Fig. 17 for an example).

Annotated TEs in *A. thaliana* are generally epigenetically silenced. However, most studies have relied on a single reference genome, making it difficult to distinguish active from inactive TEs. Our mobile-ome data identifies segregating insertions corresponding to recently active TE families, and we can also consider the age of insertions, which should be proportional to their population frequency.

We investigated silencing in sSVs by remapping the existing methylation data[53] to our genome assemblies (Methods), and the results supported our conclusions above (Supplementary Fig. 18). sSVs corresponding to TEs or TE fragments are generally highly methylated, while other categories are more variable, consistent with a subset of these sSVs corresponding to highly methylated un- or mis-annotated TEs. For sSVs corresponding to complete TEs, methylation increases with frequency, indicating that older insertions are more highly methylated. Across all categories, sSVs that are part of the graph of nestedness are highly methylated, consistent with silencing targeting the mobile-ome.

Expression patterns from existing RNA-seq data[54] agree with the methylation data: sequences in sSVs corresponding to complete TEs are barely expressed (except in pollen, where some TE expression is known to occur[55]), while the behavior of sequences in other sSVs is more variable (Supplementary Fig. 19).

## The gene-ome

To investigate the variation in the portion of the genome containing PC genes–the 'gene-ome'–with minimal reference bias, we annotated each genome independently using sequence-based gene modeling and RNA-seq data from four tissues[54] (Methods). After including 1,789 TAIR10 genes that were not re-identified in our study and filtering out low-confidence genes (Methods; Supplementary Note 6–'Details about reconciling annotations and gene filtering'), the final annotation contained 34,153 genes, with 28,138 classified as bona fide PC and 5,674 as TE (Fig. 6a and Supplementary Note 6–'Genes and TEs'). Of these, 2,661 PC and 565 TE genes were not previously annotated (for a detailed analysis of new genes, see Supplementary Note 6–'New genes').

Focusing on gene-locus presence–absence variation, we found that 13% of genes were segregating in the population of 27 accessions (Fig. 6b). This variation could reflect deletions of ancestral genes or insertions of new genes specific to *A. thaliana*. To resolve this, we used *Arabidopsis lyrata* as an outgroup (Fig. 6c). This analysis revealed a striking difference between TAIR10 genes and previously unannotated genes: while the former are generally ancestral, the latter are not (Fig. 6d). There was also a clear difference between segregating and fixed genes: as expected, most of the latter are ancestral, but the vast majority of the segregating genes are not (Fig. 6e). Thus, although we may have underestimated the fraction of ancestral genes by relying on the *A. lyrata* annotation, our results suggest that segregating ancestral deletions are rare.

Segregating genes are more common near centromeres, while fixed genes are more common in the arms (Fig. 6f; $\chi^2$ test, $P < 0.0005$ across chromosomes). Syntenic ancestral genes are also enriched in the arms, while other categories are evenly distributed across the genome. As expected, high-frequency TE insertions are more common near centromeres, consistent with stronger purifying selection in the arms[38].

New PC genes were enriched in defense and zinc-finger genes, while TAIR10 genes were enriched in housekeeping functions such as transcriptional regulation and membrane proteins (Fig. 6g; for details of functional annotation, see Supplementary Note 6–'Genes and TEs'). A similar difference was found between ancestral and non-ancestral genes (Fig. 6h), as well as between fixed and segregating genes (Extended Data Fig. 5e,f). New TE genes were strikingly more enriched for TE-function proteins than already annotated TEs (Fig. 6g), suggesting that the former are more likely to be active TEs—as expected given that they are actually segregating.

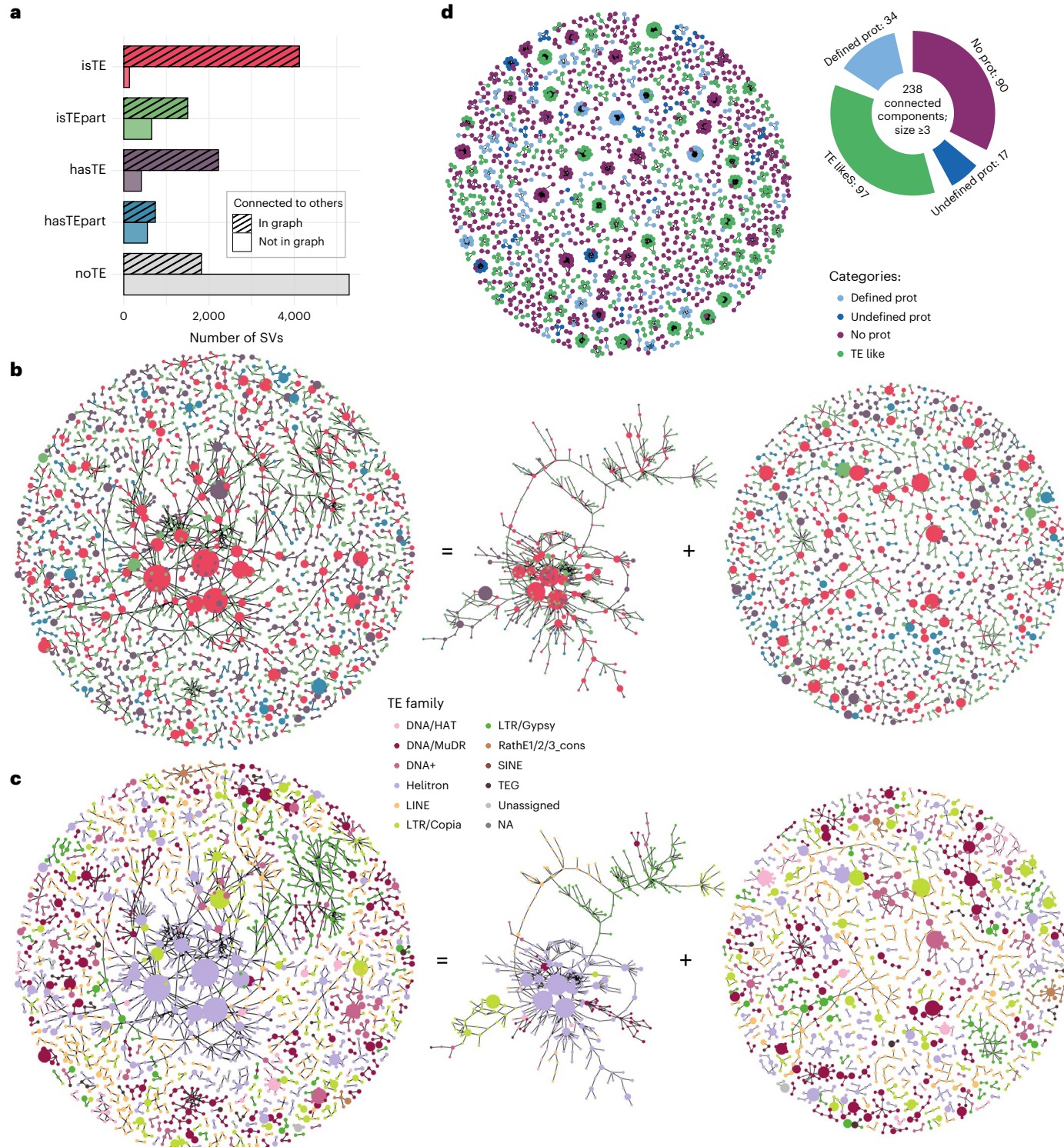

**Fig. 5 | The graph structure of sSVs. a**, The number of sSVs included (or not) in the graph of nestedness as a function of TE content. **b**, Graph illustrating the nestedness of sSVs, including only presence alleles with nonzero TE content. Each node represents a set of mutually nested sequences, using a similarity threshold of 0.85, with node size reflecting the number of included sequences. Nodes are colored by TE content as shown in **a. c**, Same graph as in **b**, but colored by TE superfamilies, as shown in the center on top. Note that the algorithm used no prior knowledge of TE superfamilies. A large, dominant component connecting all TE superfamilies is also seen in a graph built using the entire *A. thaliana* TE annotation, suggesting that our sSVs and the reference annotation have comparable properties (Extended Data Fig. 4). **d**, A graph of nestedness for sSVs without TE content, colored by open reading frame content based on protein BLAST, as shown on the right.

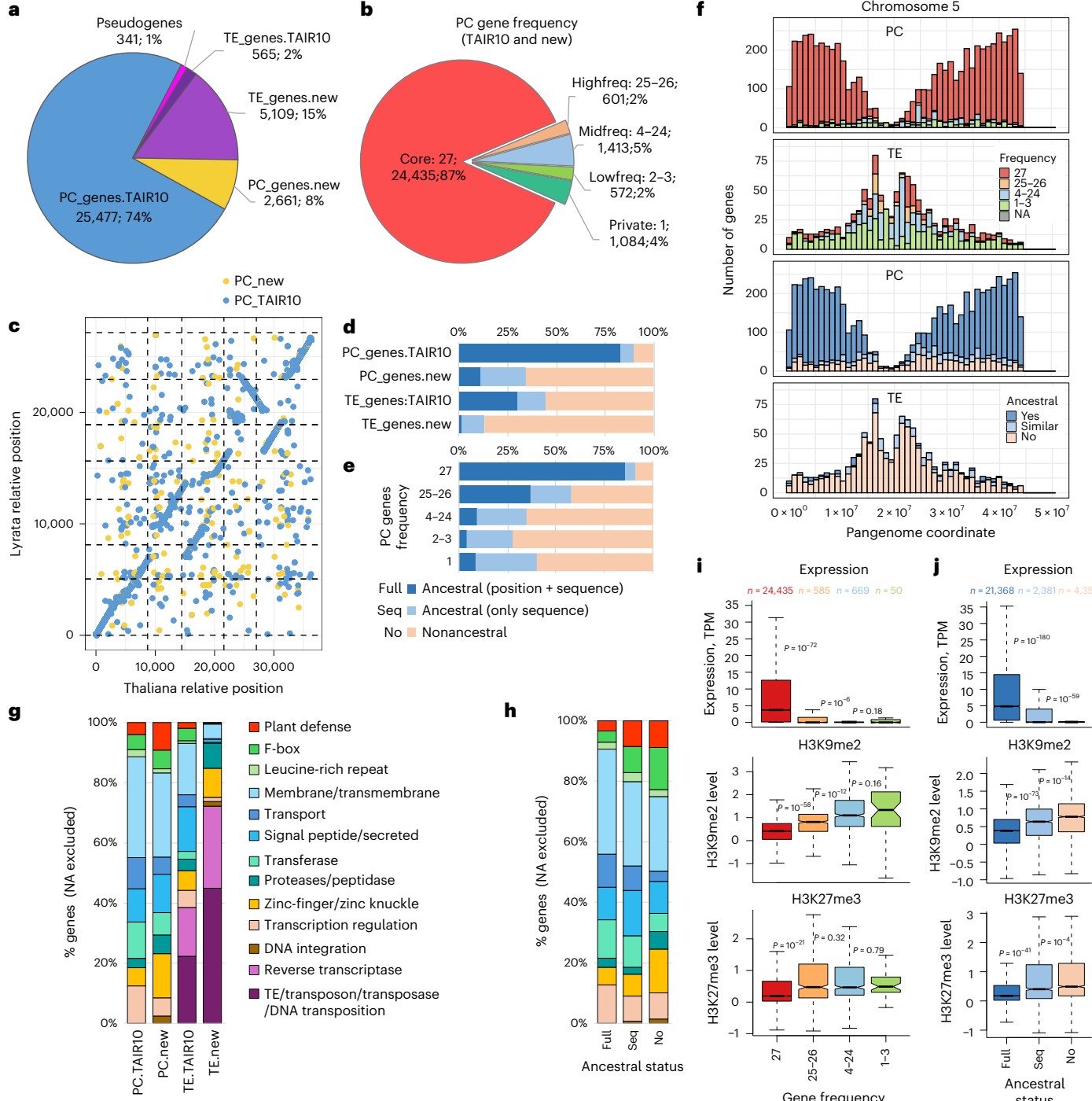

**Fig. 6 | Analysis of the gene-ome. a**, The 34,153 annotated genes categorized based on whether they correspond to the TAIR10 annotation and whether they are TEs or PC (based on sequence similarity and TE-like protein domains; Extended Data Fig. 5a). **b**, Genes categorized based on locus presence frequency across the 27 genomes. For TE presence frequency distribution and a comparison with published results[23], please see Extended Data Fig. 5b,c. **c**, Synteny comparison with *A. lyrata*. For TEs, see Extended Data Fig. 5d. **d**, Ancestral status of genes of TAIR10 and newly annotated genes ('position + sequence' means that both gene sequence and syntenic position are conserved). **e**, Ancestral status of genes by presence frequency. **f**, Distribution of genes and TEs along chromosome 5, grouped by frequency (top) and ancestral status (bottom). For all chromosomes, see Extended Data Fig. 5h,i. **g**, Functional domains of annotated genes, based on comparisons with UniProtKB. **h**, Functional domains of PC genes

grouped by ancestral status. In **g** and **h**, genes not matching a functional category have been excluded; for plots including these and breakdowns by presence frequency, see Extended Data Fig. 6e–g. **i**,**j**, Expression (in 9-leaf rosettes) and H3K9me2/H3K27me3 levels (in mature leaves) by gene frequency (**i**) and ancestral status (**j**). Data shown[54] are for accession 6024; for other accessions, as well as DNA methylation and 24-nt sRNA coverage, see Extended Data Fig. 6. Box plots show the median (center line), the 25th and 75th percentiles (box bounds) and the smallest and largest values within 1.5× the interquartile range (whiskers); outliers beyond this range are not plotted. Expression and H3K9me2/H3K27me3 were compared between frequency/ancestry gene categories using a two-sided Wilcoxon rank-sum test. The number of genes in each category is shown above the top box plot. No multiple-comparison corrections were performed.

Finally, fixed genes are far more highly expressed than segregating genes (Fig. 6i). Indeed, genes absent in only one or two accessions tend to be almost silent in the remaining accessions (Extended Data Fig. 6). Epigenetic silencing behaves similarly, with both gene-like Polycomb silencing (H3K27me3) and TE-like silencing (H3K9me2 and DNA methylation) being substantially higher for segregating genes (Fig. 6i). Likewise, nonancestral genes have reduced expression and increased silencing compared to ancestral genes (Fig. 6j and Extended Data Fig. 6).

## The pangenome

The term 'pangenome' is currently applied with a variety of meanings, from the original use in prokaryotes to describe the observation that genomes from different strains of the same species vary enormously in gene content[56], to the human genetics ambition of representing all polymorphism in a single 'pangenome graph'[15,28]. We will discuss the utility of these concepts further below—here we simply consider how the known pangenome grows with sample size and why.

All components of variation considered in this paper grow with sample size, but at different rates: the mobile-ome grows faster than the full genome and the gene-ome more slowly, consistent with the latter being under stronger purifying selection (Fig. 7a). All components increase faster than the logarithmic growth expected under neutrality in a constant population and more slowly than the linear growth expected in an exponentially growing population[57].

The growth of the pangenome is reflected in the coordinate system and is not uniform along each chromosome because most of the variation is found in centromeric regions (Fig. 7b). Already with 27 accessions, the pangenome chromosomes are 63–76% longer than the TAIR10 chromosomes.

## Missing polymorphism

As part of the 1001 Genomes Project, we previously 'resequenced' 1,135 accessions using short reads[34]. We were well aware that the data were both incomplete and error-prone: we only called SNPs and short SVs, and only an average of 84% of the reference genome was covered by short reads from any particular accession.

With our whole-genome polymorphism data, we are now in a position to assess how much variation was previously missed. In the 1001 Genomes data, a pair of accessions differed, on average, at ~440,000 SNPs. In our Pannagram alignment, the corresponding number ranges from 600,000 to 800,000 SNPs, depending on how SNPs are defined (Methods). In other words, we previously missed 25–45% of the SNP variation. In addition, WGAs of two genomes reveal, on average, ~190,000 SVs (of length <10 kb) covering a total of over ~12.5 Mb of sequence—approximately 10% of each assembled genome (Fig. 7c).

We investigated the causes of the missing SNPs by calling SNPs for our 27 genomes using the PCR-free, high-coverage short reads that were generated for this study and comparing the results to those from pairwise WGA of complete assemblies (Fig. 8a). The main reason for missing SNPs from short reads comes from not making calls in regions that are not reliably covered by reads due to mapping problems. The extent of such regions depends on the mapping parameters used; however, less conservative read mapping will generally come at the cost of higher error rates. In our test SNP-calling with PCR-free data, we were able to reduce the fraction of missing SNPs to below 20%, but our false discovery rate (FDR) was then close to 7%. Consistent with a trade-off between conservative and aggressive read mapping, bona fide SNP-calling errors, be they false positives or false negatives, were overwhelmingly due to read mismapping. Rampant pseudoheterozygosity caused by segregating duplications that are absent from the reference genome is particularly worrisome[48] (pseudoheterozygosity is easy to distinguish from residual heterozygosity in inbred lines; only one of

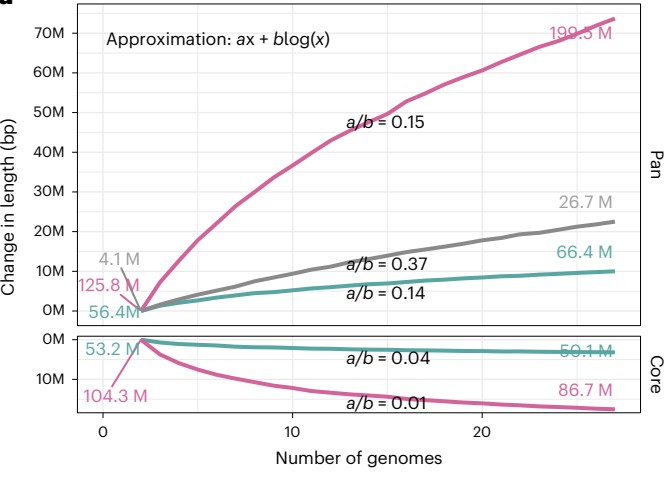

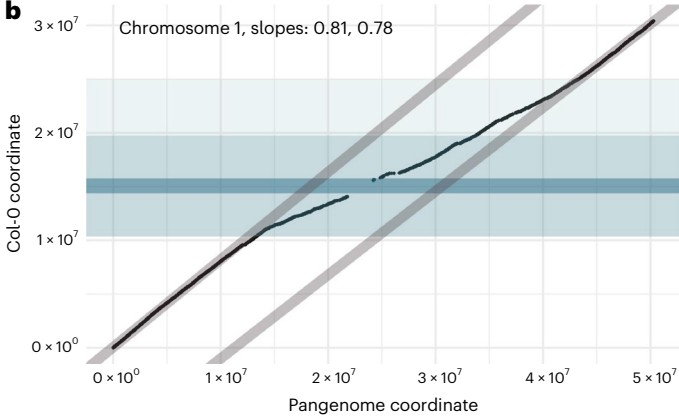

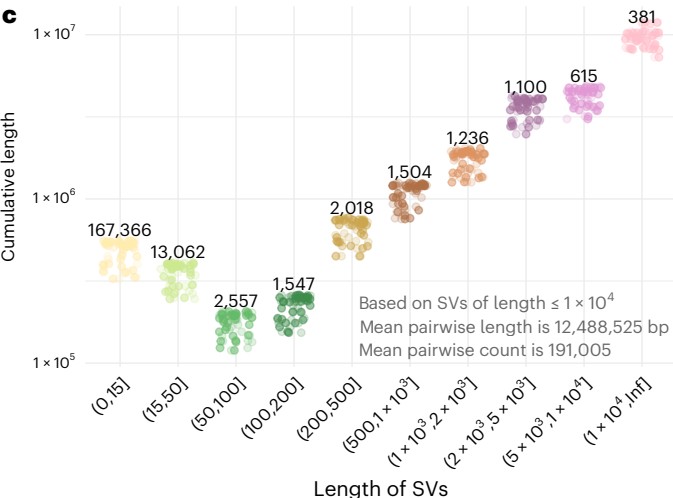

**Fig. 7 | The growth of the pangenome. a**, The dependence on sample size for the union ('pan') and intersection ('core') sequence length, separately for the full genome, the mobile-ome and the gene-ome (for saturation by sSV length and normalized views, see Extended Data Fig. 7). **b**, Pangenome versus reference genome coordinates. The pericentromeric region (light blue) shows a higher 'dilution' of the spatial coordinates due to the increased number of SVs in this region, but please note that the slope is never 1, reflecting their ubiquity. The centromeric region is dark blue (for other chromosomes, see Extended Data Fig. 7). **c**, The contribution of sSVs of different lengths to differences between pairs of genomes. Numbers are counts of sSVs. Shorter sSVs are more numerous, but longer and rarer sSVs contribute more to total length variation.

our accessions had very limited amounts of the latter). Differences in local sequence alignment between algorithms also contributed to discrepancies, whereas traditional base-calling errors had an insignificant role (Supplementary Fig. 27).

### Reference bias

Mapping short reads to a reference genome can cause reference bias; that is, the results will depend on which genome is used as a reference. Starting with SNP calling, we found that all SNP error rates depend both on the reference genome and on the relatedness between the reference and the sampled genome, often in unpredictable ways (Fig. 8b). Because many population genetic analyses rely on SNPs to estimate the relatedness between samples, this is troubling, and the consequences of nonrandomly varying levels of bias in samples for downstream analyses such as genome-wide association study or demographic inference merit further investigation.

Reference bias may also affect analysis using standard omics techniques that quantify molecular phenotypes by mapping short reads to a reference genome. We investigated this problem for transcriptome and methylome profiling by comparing the results of mapping RNA-seq and methylome profiling by comparing the results of mapping RNA-seq and bisulfite sequencing (BS-seq) reads to the TAIR10 reference as well as to the genome of the accession being studied. Expression estimates between the two approaches were strongly correlated on average (Fig. 8c), but a subset of genes diverged markedly (Fig. 8d). These were strongly enriched for copy-number variable genes (Fig. 8e). Methylome profiling was even more sensitive to the choice of genome for mapping—not surprisingly given that methylation in *A. thaliana* predominantly targets highly variable TEs[48]. A scan for differentially methylated regions in 100-bp windows across the genome revealed that profiling methylation by mapping reads to the TAIR10 reference rather than to each sample's own genome produced a large number of spurious differentially methylated regions (Supplementary Fig. 32). How serious this problem is will depend on the application, but it seems clear that it must be considered.

## Discussion

Over the last several years, population samples of more-or-less complete eukaryotic genomes have been appearing at an increasing rate, and there has been much excitement over the (variously defined) 'pangenome', especially in plants[58]. At the same time, it has become abundantly clear that a principled characterization of all the differences between individual genomes is very difficult. The problem is not a technical one, because even with perfect chromosomal sequences, we have to decide how to align them and how to interpret the differences. For complex structural variation, especially in highly diverged regions[59], this is not trivial, and there is no obvious 'gold standard' by which to evaluate algorithms. Furthermore, the ultimate reason for comparing genomes matters. If we are interested in using the pattern of polymorphism to answer questions about evolutionary history and

mutational mechanisms, we must employ models that reflect actual historical events. In contrast, if the goal is simply to develop easily usable genetic markers, it may be irrelevant whether there is any correspondence between designated variants and the molecular processes that generated them.

With these caveats in mind, we tried two different approaches: Pannagram[46], a whole-genome multiple-alignment pipeline, and PGGB[15,29], a tool primarily developed for human genomes. This is relevant because *A. thaliana* genomes have higher levels of polymorphism, stronger population structure and many recently active TEs—all of which complicate graph building and interpretation. For these reasons, as well as general convenience, we based most of our analysis on the Pannagram output.

It has long been clear that SVs contribute substantially to polymorphism[60,61]. In addition to massive variation in tandem repeat regions (Fig. 1), the readily alignable chromosome arms (Fig. 2) are highly polymorphic, with two accessions differing at ~191,000 SVs covering ~12.5 Mb on average. Although we also uncover massive amounts of new SNP variation, this still means that at least an order of magnitude more nucleotide sites are affected by SVs than by SNPs. The allele frequency distribution of SVs suggests purifying selection, as well as a mutational process involving the insertion of longer segments coupled with the deletion of shorter segments. This is consistent with TE activity, and the overwhelming majority of presence–absence variants longer than 100 bp involve annotated TE sequences. As expected in an organism with active TEs[38,62], we found thousands of examples of what appear to be recent insertions of presumably complete TEs. Many of these correspond perfectly to annotated TEs, but many do not, demonstrating that our understanding of the mobile-ome remains highly incomplete[49].

Turning to the gene-ome, we note that the term 'pangenome' was originally invented to describe the rather fluid genomes of prokaryotes[56,63]. The overall picture in *A. thaliana* is clearly very different: the gene-ome is highly conserved, with 87% of genes detected in all 27 genomes, and the number of segregating genes growing considerably more slowly with sample size than other types of variation (Fig. 7). We distinguish between two types of segregating genes: a minority with homologs in the closely related *A. lyrata* and a majority without (Fig. 6). The former, which correspond either to gene duplications or segregating deletions of ancestral genes, tend to be expressed at substantially higher levels than the latter, which often were characterized by TE-like epigenetic silencing.

Finally, we demonstrate that algorithms for SNP calling, transcriptome profiling and methylation profiling that rely on mapping short-read data to a reference genome can be highly biased—a problem that is likely to be worse in organisms with larger and more polymorphic genomes than *A. thaliana*. Additionally, outcrossing makes problems such as pseudoheterozygosity due to cryptic duplications far more difficult to detect[48,64]. While it is still impractical to completely abandon

**Fig. 8 | Read mapping and reference bias. a**, To investigate SNP-calling errors, SNPs were identified from pairwise WGA as well as by designating one genome as a reference and calling SNPs using short reads from the other (next generation sequencing (NGS); see Supplementary Note 7 for details). Numbers are averages across all accession pairs (±s.d.). From the point of view of WGA, each site can either be aligned or not. Conversely, from the point of view of NGS, each site in the reference genome can be covered by sample reads or not. Arrows pointing right refer to WGA SNPs, which we assume to be correct; arrows pointing left refer to NGS SNPs. WGA SNPs are TPs if also called by NGS, FNs if missed by NGS and uncalled if in regions not covered by NGS reads. Conversely, NGS SNPs are TPs if also found in the WGA and FPs if not. FPs come in two flavors: those that correspond to bona fide non-SNPs in the WGA and those that correspond to regions that were not aligned. Heterozygous calls in completely inbred lines are obviously FPs and are treated separately. **b**, SNP-calling error rates depend on the relationship between the reference and the sampled genome. Each

line is the regression of SNP-calling errors for a different choice of reference genome (identified by color). FNR, false negative rate. **c**, Correlation between expression- and methylation-level estimates derived from mapping reads to the TAIR10 genome and the genome of the sampled individual. Sample sizes for each data type also apply to **d** and **e**. **d**, Percentage of genes for which expression or methylation levels differ by more than 30% or 50%, respectively, when mapping reads to the TAIR10 reference rather than their 'own' genome. **e**, Enrichment of copy-number variable genes among those with discordant expression- or methylation-level estimates. Box plots show the median plus the 25th and 75th percentiles, with whiskers covering the smallest and largest values within 1.5× the interquartile range. The percentage of copy-number variable (CNV) genes was compared between wrongly and correctly estimated genes across available samples using a two-sided Wilcoxon test. TPs, true positives; FPs, false positives; FNs, false negatives.

the use of SNP calling based on short reads in favor of whole-genome polymorphism data, we note that a recent study that adopted this approach to estimate demographic parameters in *A. thaliana* reported that parameter estimates changed by two orders of magnitude[65]. Using

at least a sample of diverse, high-quality genomes for read mapping should greatly help quantify the biases.

In conclusion, we recall the prediction made in ref. [66] at the dawn of the SNP era that models would be needed to 'make sense out of

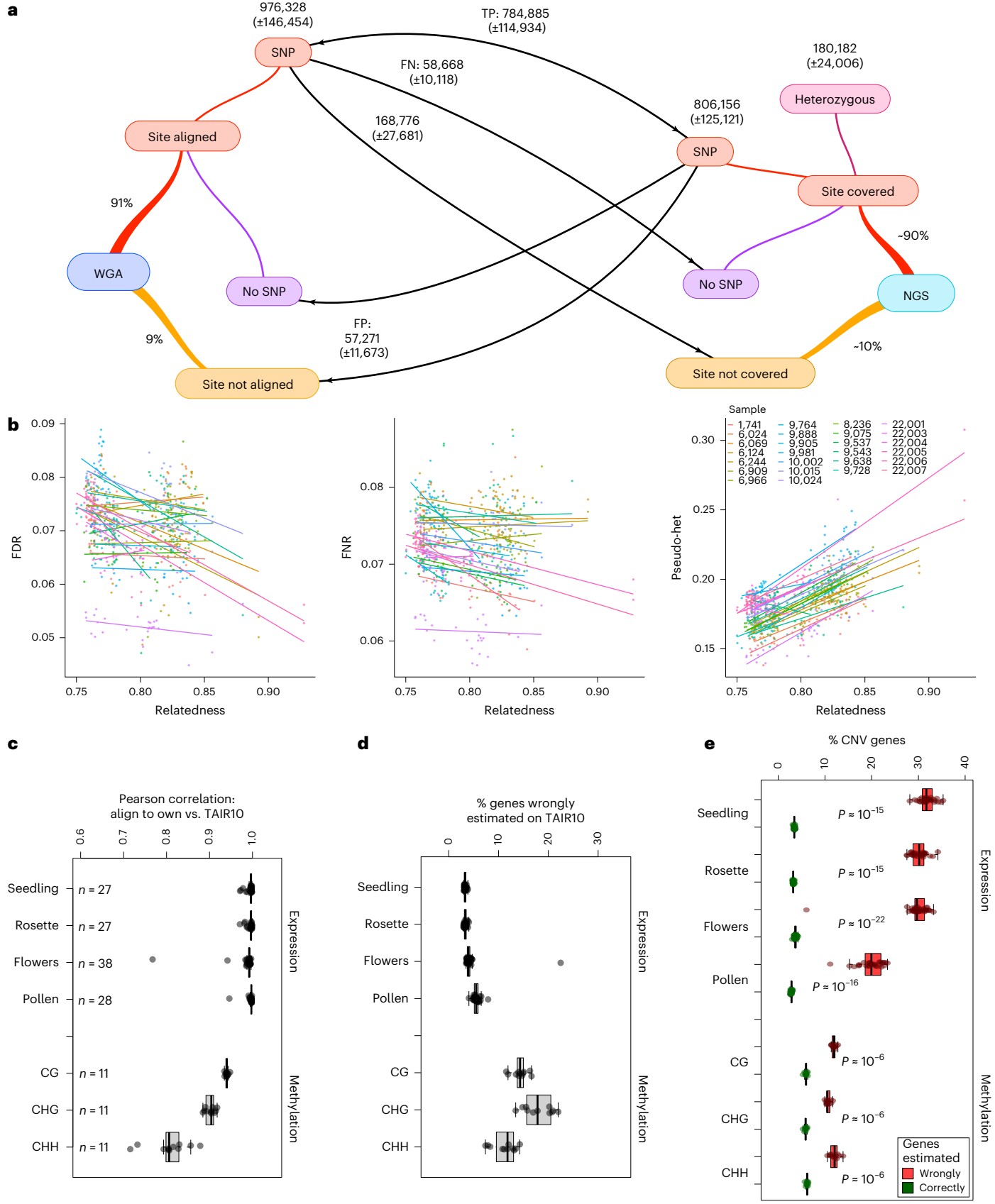

sequence'—and that this would lead to 'a rejuvenation of population genetics'. The advent of unbiased whole-genome polymorphism data will have a similar effect. Most obviously, our understanding of TE dynamics will be revolutionized by our ability to see segregating TE insertions, reflecting recent activity. While interspecific genome comparisons can reveal that bursts of activity have occurred, they lack the resolution to understand their dynamics and cannot readily distinguish active TEs from the accumulated layers of dead and decaying TE sequences that litter most genomes.

More subtly, to make sense of complex polymorphisms, we need to understand the history of mutations that gave rise to them. The problem is analogous to phylogenetic analysis, where the estimated relationship between species is used to deduce the history of complex traits—what evolved first and what evolved multiple times? In the present context, we need to estimate local coalescent trees (the so-called Ancestral Recombination Graph[67,68]) and use them to infer the sequence of mutational and recombination events that gave rise to the sampled sequences. At the same time, this analysis must be informed by a better understanding of the molecular mechanisms that cause structural variation. Although a considerable literature on phylogeny-guided statistical alignment exists[69–73], methods for doing this on a whole-genome scale, using an appropriate population genetic framework[57] and incorporating knowledge about molecular mechanisms, are missing.

In this context, it is important to remember that alignment-based methods developed for humans often do not work well in other species for a variety of reasons, including much higher levels of polymorphism and TE activity. *A. thaliana* genomes are not unusual in these respects. In species like maize, where intergenic space is essentially unalignable even between relatively closely related agricultural varieties[60], the idea of representing a whole-genome multiple alignment as a graph that captures all variation may be neither practicable nor useful[74].

## Online content

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

## Methods

### DNA extraction and sequencing

For long-read sequencing, we began with 3-week-old plants grown in soil that had been transferred to darkness for 24–48 h before harvesting to reduce the starch content. A total of 20–30 g of flash-frozen rosette tissue, pooled from individuals, was ground in liquid nitrogen with a pestle and mortar. Nuclei were isolated as described for accession Ey15-2 (ref. 75), and high-molecular-weight (HMW) DNA purified with Genomic-tips 100G (Qiagen, 10243) following the manufacturer's instructions. Ten micrograms of HMW DNA were sheared with either Megaruptor 3 (Diagenode, B06010003) or a FINE-JECT 26G×1″ needle (0.45 × 25 mm; 14-13651) to ca. 75 kb, and used as input for long-read library preparation with the SMRTbell Express Template Preparation Kit 2.0 (Pacific Biosciences, 101-693-800). These libraries were size-selected with the BluePippin system (Sage Science) with a 30 kb cutoff in a 0.75% DF Marker U1 high-pass 30–40 kb vs3 gel cassette (Biozym, BLF7510). Libraries for accessions 9981 (Angit-1; CS76366) and 10002 (TueWal-2; CS76405) were sequenced on a Sequel II system (Pacific Biosciences), and the others on a Sequel I system.

To prepare PCR-free libraries for short-read sequencing, the genomic DNA was fragmented to 250–350 bp using a Covaris S2 Focused Ultrasonicator (Covaris). The libraries were prepared according to the manufacturer's instructions with either the TruSeq DNA PCR-free kit (Illumina, 20015962) or the NxSeq AmpFREE Low DNA Library kit (Lucigen, 14000-2). In total, libraries for 89 accessions (including the main 27 for which we assembled their genomes) were sequenced in paired-end mode on a HiSeq 3000 system (Illumina).

The ultra-HMW DNA extraction and sample preparation for optical maps were performed as described[75,76] at Corteva Agriscience using the Direct Label and Stain technology (Bionano Genomics).

### Assembly

The CLR subreads were assembled with Canu v1.71 (ref. 77). Since accessions 9981 and 10002 had been sequenced at higher coverage on a Sequel II instrument, only about 200× genome coverage worth of reads were used for assembly. We performed two rounds of polishing on the resulting contigs of all assemblies—first with the CLR subreads and Arrow v2.3.2 (https://github.com/PacificBiosciences/gcpp), and then with PCR-free short reads and Pilon v1.22 (ref. 78).

For scaffolding, we generated hybrid scaffolds with optical maps for eight accessions (Supplementary Table 1) using Bionano Access v1.5 and Bionano Solve v3.6. The assembly was performed in pre-assembly mode with parameters nonhaplotype and no-CMPR-cut, without extend-split. Based on what we learned from these hybrid assemblies, we set the parameters for in silico scaffolding of the other genomes. We scaffolded contigs >150 kb with RagTag v1.1.1 (ref. 79; scaffold -q 60 -f 10000 -I 0.6 -remove-small) using the TAIR10 reference with hard-masked centromeres, rDNAs, telomeres and nuclear insertions of organelles to prevent misplacement of contigs due to reference bias[75]. All scaffolded assemblies were manually curated to specifically discard low-confidence centromere satellite-rich contigs or to invert contigs with satellite repeats at their edges, indicative of their correct orientation. These edits were implemented in the AGP files, which were converted to FASTA format with the RagTag agp2fa function[79]. To detect traces of residual heterozygosity, we aligned the original long reads to their corresponding chromosome scaffolds using pbmm2 v1.3.0 with the parameters align -sort -log-level DEBUG -preset SUBREAD -min-length 5000. Unmapped reads, as well as secondary and supplementary alignments, were filtered out using samtools v1.9 (view -b -F 2308 Chr1 Chr2 Chr3 Chr4 Chr5). The resulting BAM file was then analyzed with NucFreq v0.1 (-minobed 2) to assess genome-wide coverage of primary and secondary alleles[80]. AGP files, both before and after manual curation, as well as NucFreq plots, are available in the GitHub repository of this project.

### Repeat annotation

Repetitive elements were annotated as described[75]. We ran Repeat-Masker v4.0.9 (-cutoff 200 -nolow -gff -xsmall) using a custom library that included various consensus sequences for the CEN178 (ref. 81), 5S rDNA[82], 45S rDNA[83] and telomere repeats. We annotated tRNAs with tRNAscan-SE v2.0.6 (ref. 84) and TEs with Extensive de novo TE Annotator v1.9.7 (ref. 85; -step all -sensitive 1 -anno 1), a pipeline that combines several TE annotation tools (LTRharvest, LTR_FINDER, LTR_retriever, TIR-Learner, HelitronScanner and TEsorter)[86–93]. Finally, to understand the causes of contig breaks, we determined the type of repetitive element closest to each contig edge, considering the first 2 kb from each edge in contigs >10 kb.

### Pannagram

Pannagram is a toolkit designed for reference-free pangenome alignment, annotation and analysis, as well as for generating diagrams[46].

We represent the WGA as a matrix of corresponding positions, where rows represent accessions and columns represent homologous positions. The construction of the alignment is done in a reference-free manner (see below). However, to visualize the alignment in genome browsers, columns must be sorted in some manner, for example, to correspond to the TAIR10 sequence order. Then, columns of the pangenome are used as positions in the pangenome coordinate system.

To perform reference-free WGA, we developed a three-step pipeline. First, we use several accessions as references and build draft pairwise alignments between each and all other accessions. This process results in several reference-based matrices of corresponding positions. Next, we intersect these matrices, selecting only those columns that are present in all reference-biased matrices, which produces reliable and reference-independent correspondences. In the final step, we resolve unaligned sequences between blocks of corresponding positions using multiple sequence alignment tools. Once the reference-free alignment is complete, it can be sorted according to the desired order of accessions. In our case, we employ an alphabetical order, with the TAIR10 genome first.

For the pairwise alignments between a reference genome (not necessarily TAIR10) and another accession, the focal accession genome is divided into blocks of 5,000 bp, and each block is then mapped to the corresponding chromosomes of the reference genome using BLAST, with exactly one best hit retained for each block through this process. Next, the BLAST hits that are not in close proximity to each other in both genomes are removed. An additional BLAST search is performed to align corresponding unaligned sequences between remaining hits.

To resolve any unaligned blocks after the reference-randomization procedure, MAFFT[94] is used. Blocks longer than 30 kb cannot be aligned within a reasonable time using MAFFT, so they are considered to be highly diverged. We found the final unaligned regions to be primarily associated with centromeric regions, rDNA clusters, telomeres and complex regions of multiple and long insertions and deletions, which are regions that are not of primary interest in this paper.

Given the WGA, SNPs can simply be output as sequence differences. However, sequence differences can arise from ambiguities in local alignment and do not necessarily correspond to SNPs (Supplementary Note 7). If we consider all sequence differences as SNPs, a pair of accessions differs at over 800,000 positions on average; however, if we restrict ourselves to isolated sequence differences, the number shrinks to 600,000.

### Pangenome graph

**Graph construction.** We constructed genome graphs for each of the five chromosomes using the PGGB pipeline[29]. First, we prepared the assemblies by splitting them into chromosomes and removing all unplaced contigs. To enforce linearity for simpler analysis and comparison, we used a modified version of accession 22001 with the genome rearranged to a consensus pan-genomic order (suffix: 'f'). We added

the TAIR10 reference genome to the graph to enable anchoring and presentation of results in a reference framework.

We executed the PGGB pipeline (downloaded on 25 January 2024) with the following parameters: -s 10000 -p 90 -n 27. PGGB consists of the following three methods: an all-against-all alignment with wfmash (v0.12.4-5-g0b191bb), graph induction using seqwish (v0.7.9-2-gf44b402) and two rounds of pangenome ordering (odgi v0.8.3-26-gbc7742ed) followed by normalization with smoothxg (v0.7.2-11-g9970e0d). The graph was used for analyzing the pangenome and synteny, as well as detecting variation using vg deconstruct[95].

**Similarity.** We exploited graph properties to classify different levels of similarity between genomes. Nodes traversed in all accessions are labeled as core, nodes traversed in only one accession are private and all other nodes (>1 and <28 traversals) are shell (soft). Please note that nodes can be traversed multiple times by the same genome, which affects the total number of core nodes. Because each node contains a specific sequence, node count can easily be converted to the actual amount of sequence and respective genomic location.

**Synteny windows.** Every node in the graph can be translated to its exact position for each path. This direct connection allows us to create sliding window approaches for each sample/path using graph-based statistics. Here we used nonoverlapping windows of 300 kb and calculated the average similarity (see above) of these regions. This was performed for each graph and path independently and the results were represented in a heat map.

**Saturation analysis.** A saturation analysis was performed using a bootstrapping approach. In each iteration, we removed a specific number of paths from our graph and performed the same pangenome categorization as above ('Similarity'). In addition, we added the total pangenome, which describes the total amount of sequence (core + shell + private sequence). We performed 20 different (unique) combinations for each size (number of genomes).

**Deconstructing the graph.** To achieve full insights into graph variation and cover all bubbles in the graph, VG deconstruct was run multiple times with each accession reference path once (vg deconstruct -a -e). After, the reported VCF (v1.54.0 'Parafada') files were converted to a BED file with all important information provided. In addition, each chromosome was merged, and the genotype information was concluded and added. Bubbles were identified by the start and end positions, and all traversals within these bubbles were also reported. Scripts can be found in the repository.

sSVs and cSVs in the graphs were defined as follows:

- All SVs represent indels, having one very small traversal (deletion) and a large one containing the SV sequence (insertion).
- Bubbles were identified as sSVs if the bubble was shared by all accessions in the graph (here 28), and as cSVs if not. Traversals covering the insertion are at least 15 bp long and must exhibit high similarity (95% sequence). The deletion part of the bubble should be small, at most 5% of the length of the inserted sequence.
- Most cSVs correspond to bubbles that have a complex structure and/or are sub-bubbles of larger bubbles.

**General pangenome.** To perform a reference-free pangenome analysis, we used genome graphs built separately for each chromosome. The complete graph contains 18.3 million nodes and 20.9 million edges, with a total size of 225 Mb, and has a mean compression rate of 6.75% across all chromosomes. Similar to other genome-wide analyses in this study, the large-scale reciprocal translocation in accession 22001 was masked to maximize linearity and increase resolution in the variation graph.

## The mobile-ome
The mobile-ome refers to the collection of insertions and deletions that are likely to have occurred recently and are therefore not fixed in our sample. We hypothesize that each mobile event results in an SV, specifically a presence–absence polymorphism at the location of the insertion or deletion. Consequently, our initial approach involves extracting all presence–absence SVs and systematically decomposing them step by step. To distinguish between simple bi-allelic presence–absence polymorphisms (indels) and cSV, we analyzed the lengths of alleles within the SVs. We distinguish two types based on the similarity threshold $s$, with $s = 0.9$ in our case. We consider a simple indel as one that contains alleles of two length types—those that are shorter than $(1 − s)$ of the SV length (absence allele) and those that are longer than $s$ of the SV length (presence allele). The distinction between simple and complex presence–absence polymorphisms is partially a computational construct to filter SVs and simplify further analysis. Simple indels and complex presence–absence polymorphisms form a continuum, and by relaxing the similarity threshold ($s < 0.9$, in our case), some complex SVs become classified as simple indels. Additionally, there is a natural bias towards complex presence–absence polymorphisms. Consider a scenario with a simple presence–absence polymorphism where an indel occurs within the presence allele. If the presence allele was initially observed in only one accession, then the new event does not reclassify the initial region as not belonging to the simple presence–absence polymorphisms category. However, if the presence allele was observed in multiple accessions, the new event is likely to be reclassified as complex presence–absence polymorphisms. To simplify and clarify the analysis, we considered only the simple polymorphisms. To determine the known portion of the mobile-ome within indels, we conducted BLAST searches using pangenome consensus sequences of indels against known *A. thaliana* TEs, as well as against themselves. The indels that exhibited some similarity to known TEs were divided into the following groups: is complete—significant similarity to known TEs and can be classified as TEs themselves; contains complete—contained regions with similarity to known TEs, but also additional sequences; is fragment—contained only partially sequenced with similarity to known TEs; and contains fragment—partial coverage by BLAST hits of TE segments, but also additional sequences unrelated to known TEs.

We consider all these indels as parts of the mobile-ome. Indels without similarity to known TEs but showing nested similarities within the indel data set (where one sequence is a subsequence of another) were considered as potential candidates for new mobile-ome elements. To investigate their potential function, we obtained all six open reading frames within each of these indels. From each translated sequence, we selected either all continuous stretches without stop codons that were longer than 100 codons or the longest stretch that exceeded 30 codons without a stop. Subsequently, we performed a BLAST search using the obtained amino acid sequences against the NCBI protein database and classified the potential proteins into four categories. If the BLAST results for an sSV contained keywords related to TE, we assigned the sequence to the TE-like category. These keywords were 'transcriptase', 'reverse', 'transpos', 'gag-', 'pol-', 'integrase', 'gag/pol', 'gagpol', 'retrovirus', 'RNA-directed DNA polymerase' and 'RNA-dependent DNA polymerase'. sSVs that only had BLAST hits with descriptions such as 'hypothetical protein', 'unnamed protein product', 'uncharacterized protein', 'predicted protein', 'PREDICTED:', 'putative protein' and 'unknown' were categorized as 'undefined proteins'. Indels without any BLAST hits were classified as 'no protein'. In all other cases, sSV was categorized as a 'defined protein.'

## Gene annotation
**Preliminary annotation.** Gene annotation was mainly based on Augustus (v3.3.3)[96]. Augustus-predicted gene models were trained using parameters obtained from 'hints' from three different sources. First, we ran BUSCO (v4.0.1)[97] with -mgenome option. Second, the *A. thaliana*

reference gene annotation was projected onto each genome using Liftoff[98] with the -exclude_partial and -copies options. Third, the RNA-seq data for each accession were used—wiggle hints were generated using bam2wig and wig2hints, and EST hints were generated using bam2hints (all three tools provided by Augustus). Augustus was run with the following nondefault parameters:

```
–softmasking 1
–species=BUSCO_retraining
–gff3=on
–extrinsicCfgFile=Custom_Config
–hintsfile=Liftoff_hints
```

For every accession, the GFF3 output of Augustus was run through the Augustus-provided tool getAnno.pl to translate gene annotations into protein sequences. Finally, for each annotation, the Augustus output was combined and evaluated using augustus_GFF3_to_EVM_GFF3.pl (provided by EVidenceModeler[99]).

In addition to the Augustus-generated annotations, we used two types of independent evidence for gene models: from the SNAP de novo annotation tool[100] and Cufflinks transcriptome assemblies[101]. Annotations produced by Augustus, SNAP and Cufflinks were combined and then subdivided into 1-Mb windows with 1-kb overlap using partition_EVM_input.pl (provided by EVidenceModeler). We ran EVidenceModeler with annotation GFF files, the assembly fasta file, the partitions and a weight matrix. We chose weights for each input based on their ability to recreate the Araport11 gene annotation. Running EVidenceModeler produced the final annotation compilation for each accession. We retained only the longest isoform for each gene using gffread[102].

**Reconciling annotations.** To enable comparison between the independent annotations, we used the pangenome coordinate system, reconciling discrepancies using majority voting (Supplementary Note 6—'Details about reconciling annotations and gene filtering'). Additionally, we compared the sequences of each gene across different accessions. If a gene showed significant variation because it was located in regions heavily influenced by SVs, we excluded it from the analysis. In total, 3,438 genes in our annotation were the result of splitting preliminary annotations and 1,020 were the result of merges. Lastly, we added 1,789 TAIR10 genes that had not been detected by our annotation pipeline (the likely reason for which is that our RNA-seq data only covered four tissues/stages) to our annotation. For these genes, the same pangenome coordinate approach was used to map the TAIR10 annotation of the 1,789 added genes into their annotations in other genomes. Our approach generated a total of 34,153 putative genes. For the details of annotation reconciliation and filtering, see Supplementary Note 6—'Details about reconciling annotations and gene filtering'.

**Ancestry analysis.** All PC sequences from all accessions were compared using DIAMOND's blastp module[103] (version 2.0.11) against the *A. lyrata* MN47 proteome (version 2, GenBank: GCA_944990045.1), and the best hit was considered as the *A. lyrata* homolog. To avoid bias due to mis-annotated genes in the *A. lyrata* proteome, we further applied Liftoff v1.63 (ref. [98]) to annotate all *A. thaliana* genes from all accessions on *A. lyrata* MN47 (v2, https://doi.org/10.6084/m9.figshare.22285444.v1) and *A. lyrata* NT1 (v2, https://doi.org/10.6084/m9.figshare.22293196.v1) assemblies. Next, each annotation group from *A. thaliana* was assigned to the *A. lyrata* homolog (by LiftOff or proteome similarity) that was common to at least 50% of its members, sharing at least 80% identity and covering at least 80% of the *A. thaliana* coding sequence. *A. thaliana* annotation groups were defined to be ancestrally relative to the *A. lyrata* gene if they were part of a colinear segment of at least two genes. To that end, all *A. thaliana* genes were ordered according to their relative position in the pangenome coordinate system. Each pair of consecutive genes in *A. thaliana* was assigned to the same colinear segment as its homologs in *A. lyrata* if the homologs were separated by fewer than six genes. The ancestral state was defined as 'similar' for cases where the genes from *A. lyrata* and *A. thaliana* were not part of the same colinear segment but shared at least 80% sequence identity over at least 80% of the length *A. thaliana* gene. Further details are available in Supplementary Note 6—under 'Genes and TEs' for TE analysis and 'New genes' for the origin of new genes.

**Expression analysis**
**RNA-seq read mapping and gene expression calculation.** Raw RNA-seq reads from 7-day-old seedlings, 9-leaf rosettes, flowers and pollen[54] were aligned either to the TAIR10 reference genome or the corresponding accession accession genome using STAR v2.7.1 (ref. [104]) with the following custom options:

```
–alignIntronMax 6000
–alignMatesGapMax 6000
–outFilterIntronMotifs RemoveNoncanonical
–outFilterMismatchNoverReadLmax 0.1
–outFilterMismatchNmax 999
–outFilterMismatchNoverLmax 0.3
–outFilterMultimapNmax 1
–alignSJoverhangMin 8
–outSAMattributes NH HI AS nM NM MD jM jI XS
```

Read alignment statistics are provided in Supplementary Table 4. Expression levels were assessed using featurecounts from Subread v2.0.1 (ref. [105]) on each RNA-seq sample with either the TAIR10 gene annotation or the accession-specific annotations from this study. The entire locus, including exons and introns, was used for expression estimates. Expression levels were normalized by calculating TPMs, which represent the number of read counts divided by the gene length in kilobases, and then dividing the total number of counts per kilobase for all genes by 1 million.

**Mapping to TAIR10 versus the own genome.** To determine whether the gene expression calculation was consistent between RNA-seq mapping in TAIR10 versus accession-specific genomes, we focused on the annotation groups with a one-to-one correspondence with an Araport11 gene. For each RNA-seq sample, we obtained the Pearson's correlation coefficient between the number of exonic counts obtained from TAIR10 mapping and accession-specific mapping. We also determined the number of genes that were correctly or wrongly estimated using TAIR10 mapping. We called a gene 'wrong' if the counts in TAIR10 and the counts in its own genome differed by more than 30% (Ncounts_min/Ncounts_max ≤ 0.7). Only genes with at least six counts in either calculation were analyzed.

**Chromatin immunoprecipitation followed by sequencing analysis**
We used chromatin immunoprecipitation followed by sequencing (ChIP–seq) data from 6 accessions and sRNA-seq data from 14 accessions[54]. We used STAR[104] to map ChIP–seq reads with these nondefault options:

```
–alignIntronMax 5
–outFilterMismatchNmax 10
–outFilterMultimapNmax 1
–alignEndsType EndToEnd
–twopassMode Basic
```

The ChIP–seq data were $\log_2$-normalized to input using bamCompare (deeptools package[106]) using

```
–operation log2
–effectiveGenomeSize 119481543
–ignoreDuplicates
–outFileFormat bedgraph
```

The ChIP–seq coverage was estimated using bedtools map-mean[107]. The ChIP–seq coverage was further normalized to obtain value range similarity across accessions. For this, we applied quantile-normalization using an *R* function:

```
function(x) { (x-quantile(x,.20)) / (quantile(x,.80)
 - quantile(x,.20)) }
```

which equalized the 20% and 80% quantile values of each ChIP–seq sample. After quantile-normalization, the replicated samples were averaged.

### sRNA-seq analysis
We used sRNA-seq data for 14 accessions[54]. To process the sRNA-seq data, we trimmed the reads using cutadapt[108]: cutadapt -a AACT-GTAGGCACCATCAAT –minimum-length 18. We then used STAR[104] with the following nondefault options to map sRNA-seq reads to the corresponding genome:

```
–runRNGseed 12345
–alignEndsType Extend5pOfRead1
–alignIntronMax 5000 –alignSJDBoverhangMin 1
–outReadsUnmapped Fastx –outSAMmultNmax 100
–outSAMprimaryFlag AllBestScore
–outSAMattributes NH HI AS nM NM MD jM jI XS
–outFilterMultimapNmax 10
–outFilterMatchNmin 16
–outFilterMatchNminOverLread 0.66
–outFilterMismatchNmax 2
–outFilterMismatchNoverReadLmax 0.05
–outFilterIntronMotifs RemoveNoncanonicalUnannotated
–twopassMode None
```

We extracted 24-nt reads, calculated read coverage for each position of the genome using genomeCoverageBed (bedtools v.2.27.1), normalized it by the total number of uniquely mapped reads in each sample, and calculated 24-nt sRNA coverage for each locus of interest using bedtools map -mean function.

### DNA methylation analysis
To estimate DNA methylation levels, we used published BS-seq data for 12 accessions[53]. After trimming with TrimGalore (https://github.com/FelixKrueger/TrimGalore) with –clip_r1 10 –clip_r2 15 –three_prime_clip_r1 10 –three_prime_clip_r2 10, reads for each accession were mapped to its corresponding genome with Bismark[109] with –score_min L,0,-0.5 for a relaxed mismatch threshold and the –un –ambiguous parameters to obtain additional unmapped and multiply-mapping reads. Methylation was called as described[110]. CG, CHG and CHH methylation levels for genes and SVs in each accession were then calculated for each gene by focusing on all Cs in the specific context within the gene and calculating the ratio between the total number of methylated and unmethylated reads across all sites.

**Mapping to TAIR10 versus own genome.** To estimate reference bias, we mapped BS-seq data for all accessions to the TAIR10 genome and performed CG, CHG and CHH methylation level estimation in the same way as for own genomes. We then focused on annotation groups with a one-to-one correspondence with an Araport11 gene (the current annotation of the TAIR10 genome). We calculated Pearson's correlation coefficient between the methylation level estimates obtained from TAIR10 mapping and accession-genome mapping. We also estimated the number of genes that were correctly or wrongly estimated using TAIR10 mapping. For each methylation context, we called a gene 'wrongly estimated' if the methylation level in TAIR10 and the own genome differed by more than 50% (methlevel_min/

methlevel_max ≤ 0.5). For a more refined analysis of reference bias, see Supplementary Note 8.

### Reporting summary
Further information on research design is available in the Nature Portfolio Reporting Summary linked to this article.

## Data availability
Raw sequencing data (PacBio CLR and Illumina PCR-free short reads) and genome assemblies have been deposited in the European Nucleotide Archive (ENA) under project accession PRJEB73474 (ERP158243). Illumina PCR-free short reads for 61 additional accessions used to investigate the contribution of satellite repeats can be accessed under project accession PRJEB73476 (ERP158245). BS-seq data from mature leaves of 14-leaf rosettes are from the 1001 Genomes Project[53] and are available under GEO accession GSE43857. ChIP-, RNA-seq and sRNA-seq data were likewise previously published[54]: ChIP–seq data from 14-leaf rosettes are available under GEO accession GSE226682; RNA-seq data from seedlings, 9-leaf rosettes, flowers and pollen are available under GEO accession GSE226691; and sRNA-seq data from flowers are available under the GEO accession GSE224571. Several widely used public databases were used in the analyses: NCBI Protein Blast, UniProtKB (version 2024_06), and the TAIR10 genome annotation. Maps were generated using public domain data from the Natural Earth project via the R package rnaturalearth. See also https://1001genomes.org/, where a collection of accession-based JBrowse2 genome viewers and a pan-genome JBrowse2 browser are available, along with other relevant information.

## Code availability
All scripts, including methods for analyzing the pangenome graph, can be found in an ad hoc GitHub repository (https://github.com/Gregor-Mendel-Institute/1001Gplus_paper; archived at https://doi.org/10.5281/zenodo.15790915)[111]. The Pannagram toolkit[46] can be found in a separate repository (https://github.com/iganna/pannagram; v1.1 archived at https://doi.org/10.5281/zenodo.15791785)[112], and the same is true for the automated PacBio long-read genome assembly pipeline (https://github.com/weigelworld/auto-asm; archived at https://doi.org/10.5281/zenodo.15775624)[113].

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

## Acknowledgements

We thank Z. Bao, A. Duque and G. Ofir for discussion and comments on the paper. E. Grigoreva supplied the polymorphism data for Fig. 2c. M.N. wrote much of this paper while on a sabbatical in M. Przeworski's lab at Columbia University and thanks all members of her lab, as well as the labs of G. Sella and P. Andolfatto, for their hospitality and feedback. This work was funded by the Deutsche Forschungsgemeinschaft (DFG), the Austrian Science Fund (FWF) and the Biotechnology and Biological Sciences Research Council (BBSRC) through ERA-CAPS Project 1001GenomesPlus (BBSRC BB/S004661/1 to P.K., D.W. and M.N.), the Austrian Academy of Sciences (to M.N.), the Max Planck Society (to D.W.) and the European Union's Framework Programme for Research and Innovation Horizon 2020 (2014–2020) under the Marie Curie Skłodowska Grant Agreement 847548 (to H.-J.L.), as well as ERC Advanced Grant EPICLINES (to M.N.).

## Author contributions

P.K., D.W. and M.N. conceived and supervised the project. P.J.F., S.N., A.H. and Y.-L.G. contributed materials. F.A.R., A.E.K., M.C., A.M.M., V.L., V.N., I.R., C.L. and F.B. generated data. F.A.R., H.A., B.J. and T.W. assembled the genomes. A.A.I., H.A., M.C. and C.K. annotated the genomes. H.-J.L. and A.E.K. analyzed expression, epigenetics and short-read mapping bias. A.A.I. and S.V. performed the SV analysis. A.A.I. analyzed the mobile-ome. A.A.I. and A.D.B. analyzed the pangenome alignment. A.A.I., H.A., A.E.K. and V.V. analyzed the gene-ome. J.F. developed the genome browser. I.B. provided computational support. A.A.I., S.V., F.A.R., H.-J.L., H.A., A.E.K., D.W. and M.N. wrote the paper.

## Funding

## Competing interests

D.W. holds equity in Computomics, which advises plant breeders. D.W. also consults for KWS SE, a plant breeder and seed producer with activities throughout the world. J.F. is an employee of Tropic TI, Lda. The other authors declare no competing interests.

## Additional information

**Extended data** is available for this paper at https://doi.org/10.1038/s41588-025-02293-0.

**Correspondence and requests for materials** should be addressed to Detlef Weigel or Magnus Nordborg.

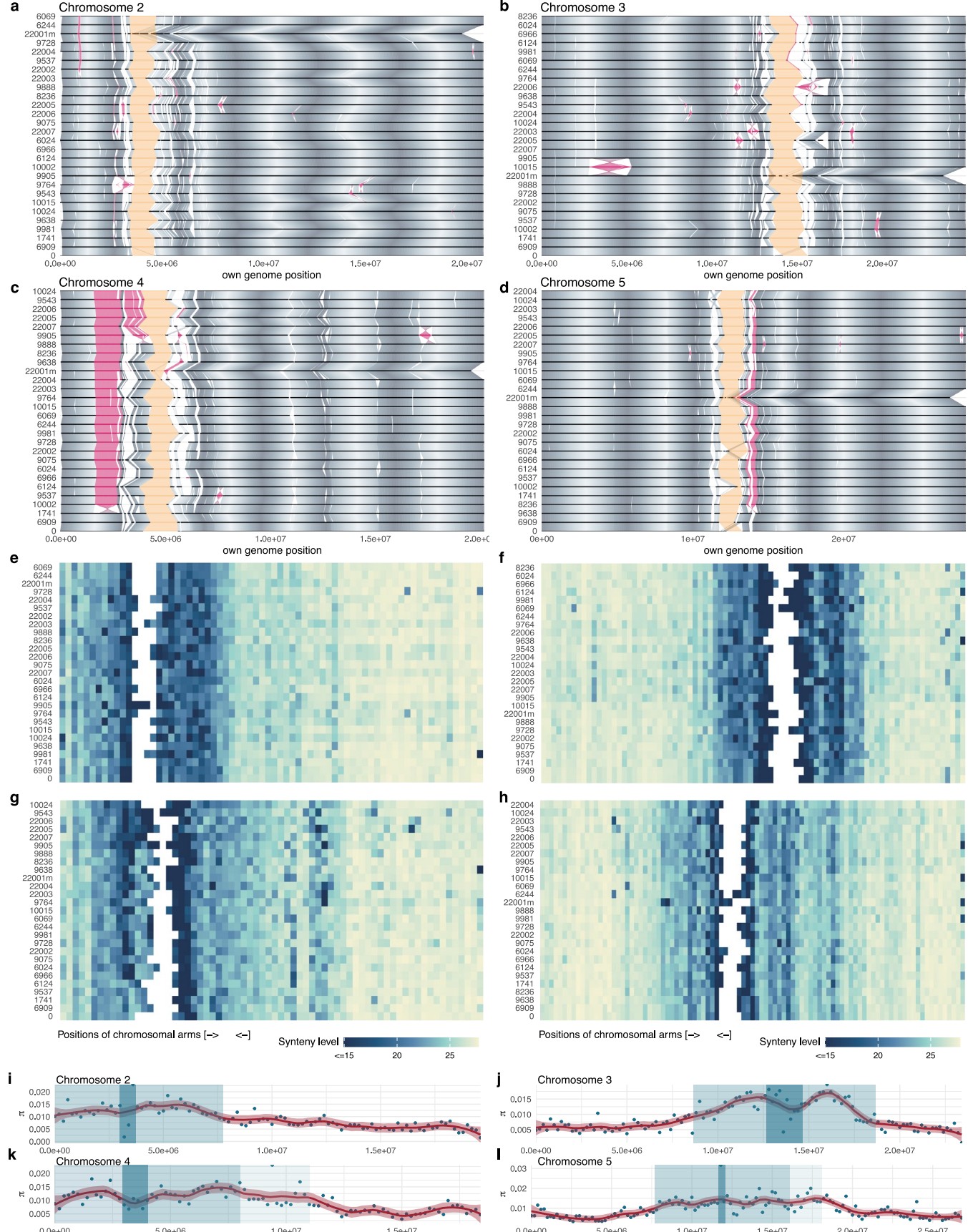

**Extended Data Fig. 1 | The pattern of polymorphism on chromosomes 2–5. a–d**, Whole-genome alignments. **e–h**, Density of PGGB graph bubbles. **i–l**, Nucleotide diversity. For chromosome 1 and further explanation, see Fig. 2a–c.

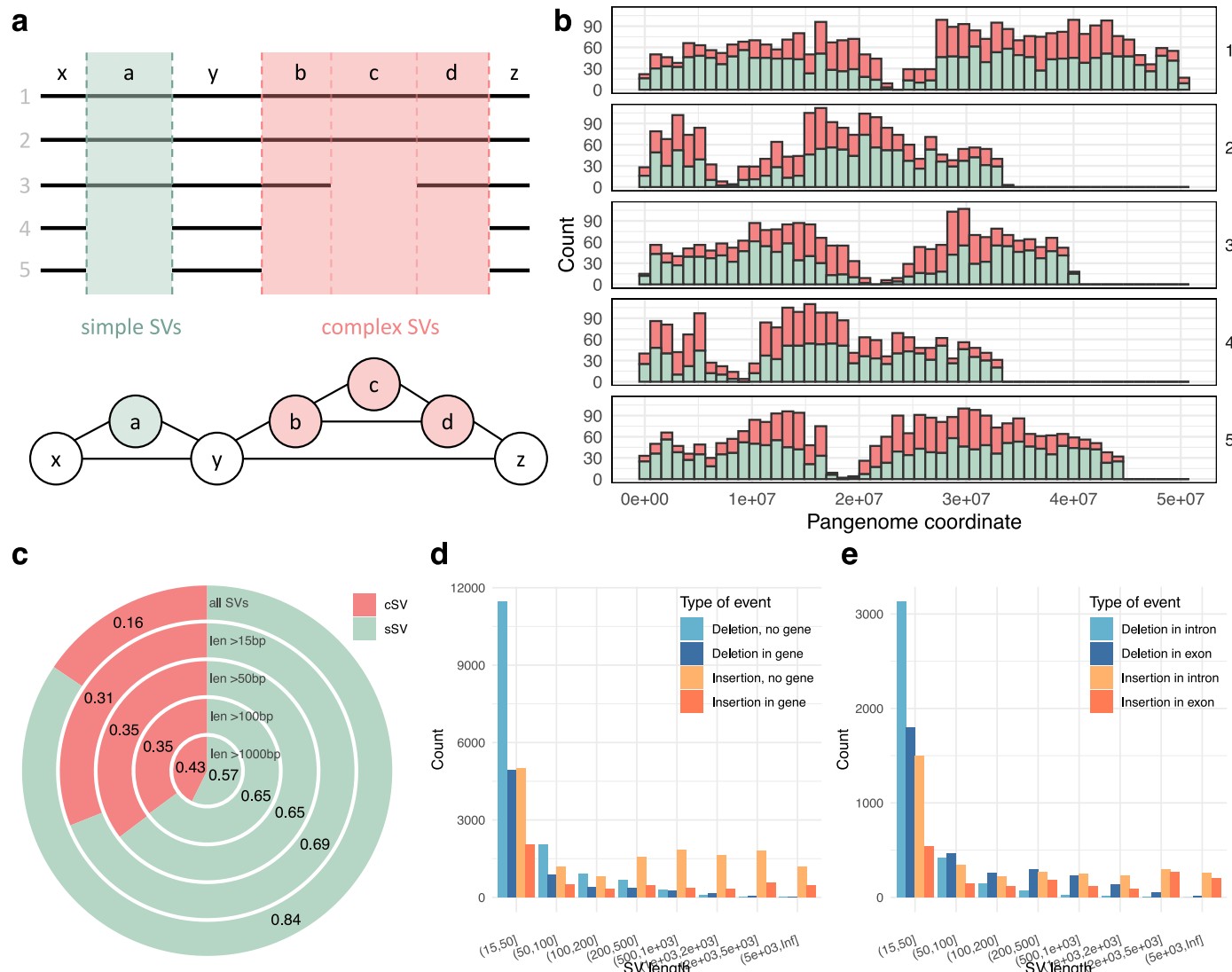

**Extended Data Fig. 2 | SV type and gene-relative location. a**, Cartoons illustrating our classification of length variants into sSVs and cSVs in the whole-genome alignment and graph representations. **b**, The distribution of sSVs and cSVs across chromosomes—cSVs are more common in pericentromeric regions. **c**, Fraction of sSVs and cSVs by length. The prevalence of short sSVs is at least partly explained by the low probability of the multiple events required for cSVs to occur in short DNA segments. **d**, The number of putative insertions (presence allele in 1–3 accessions) and deletions (absence allele in 1–3 accessions) of varying lengths in genes and intergenic regions. **e**, Same for exons and introns.

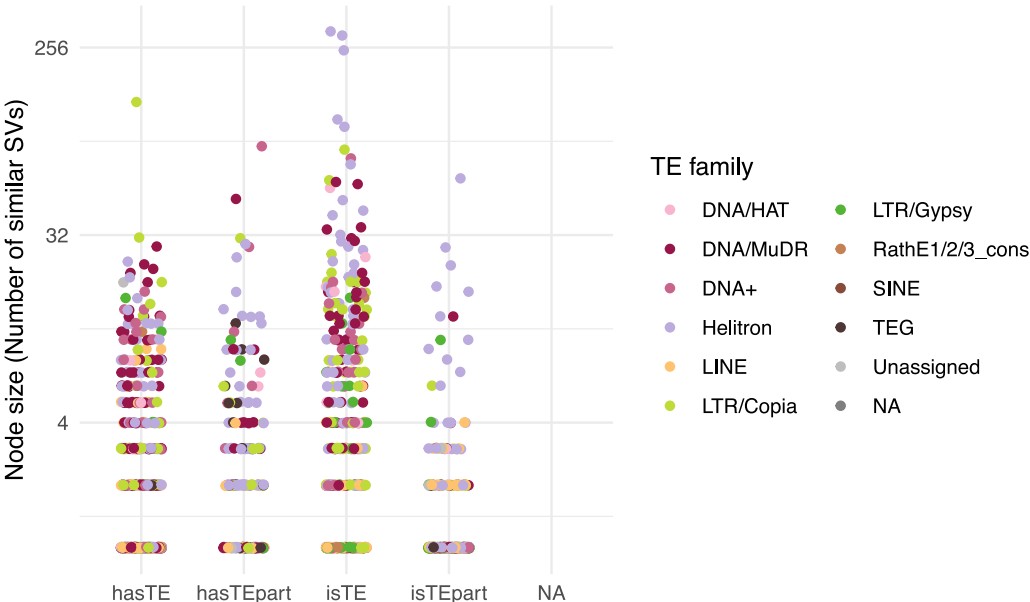

**Extended Data Fig. 3 | Size distribution of graph nodes by TE-content.** Large nodes are found not only for presence alleles that correspond to complete annotated TE but also for other categories, demonstrating that these are also part of the mobile-ome.

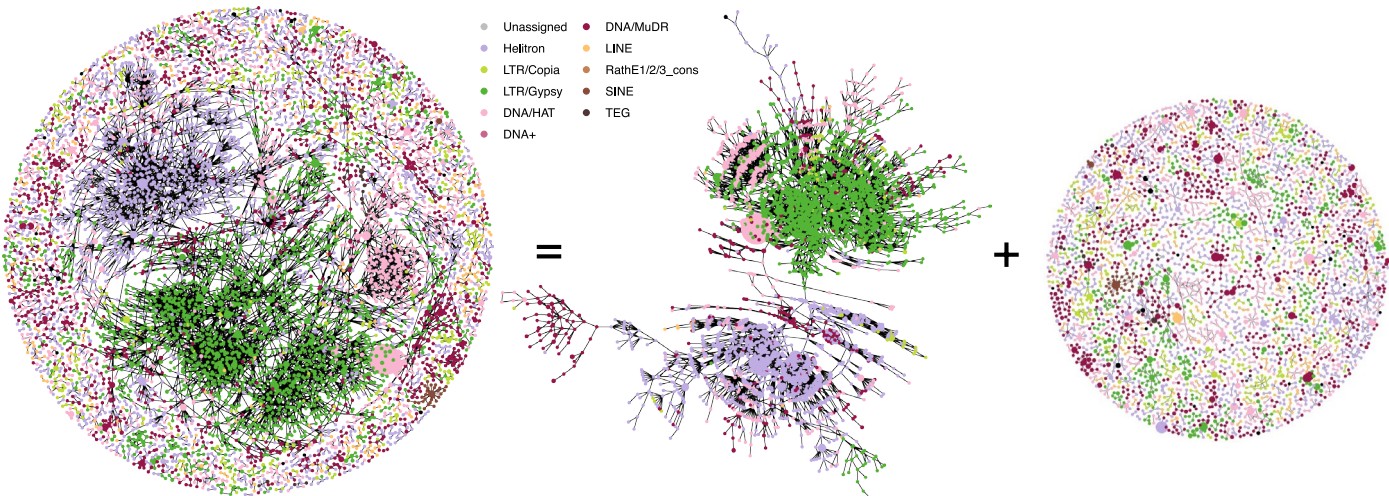

**Extended Data Fig. 4 | The graph of the nestedness of existing *A. thaliana* TE annotations.** Each node is a cluster of similar sequences (with length and identity thresholds of 0.85), where the size of the node indicates its relative abundance. The graph can be decomposed into one dominant connected component and several smaller ones, as shown on the right.

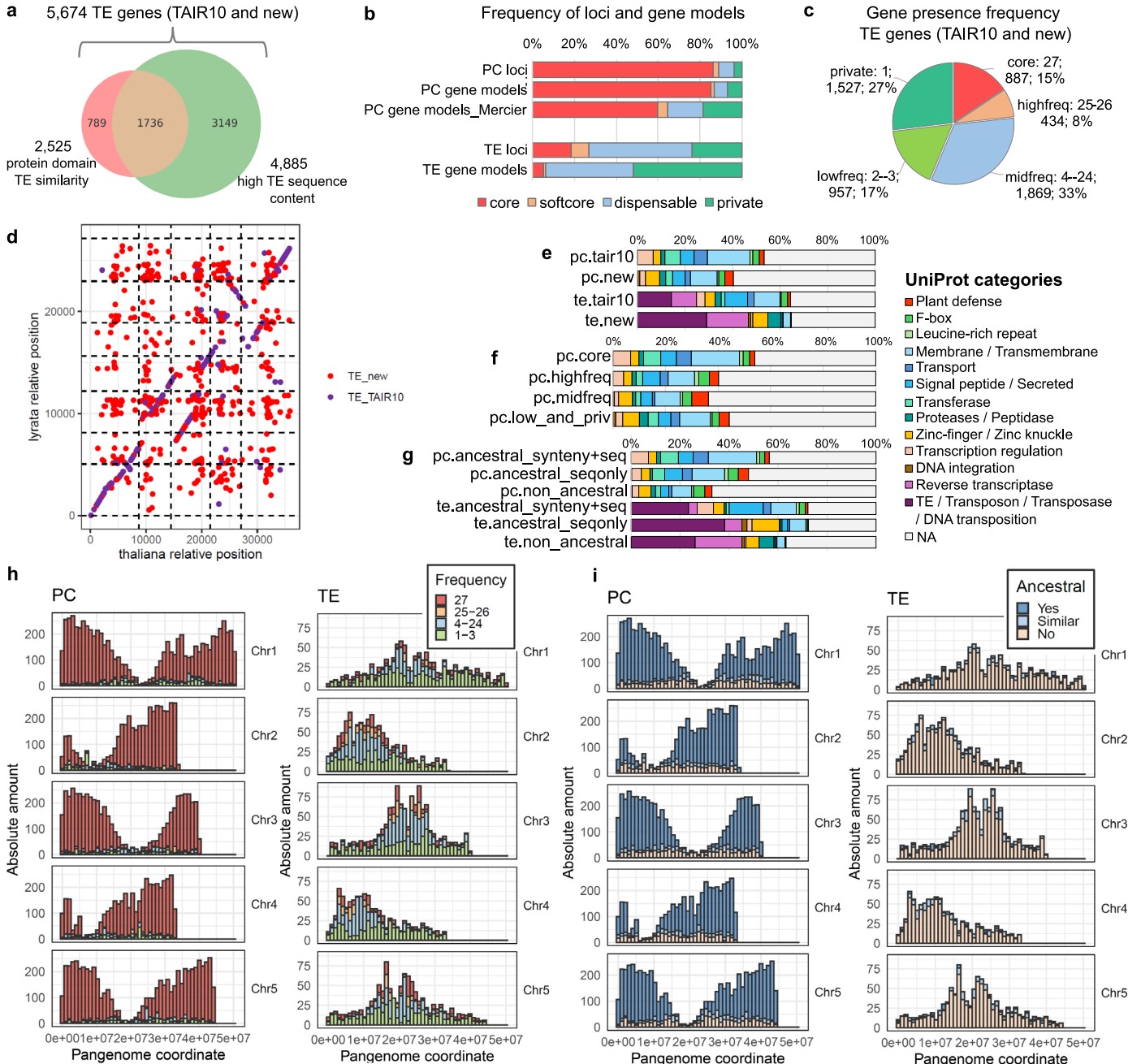

**Extended Data Fig. 5 | Details of gene-ome analysis. a**, TE genes were defined as the union of two sets of genes: (1) those with >50% annotated TE sequence in at least one accession, and (2) those with ORFs similar to proteins known to be involved in TE function. **b**, Comparison of gene presence variation in this study to a recent analysis in 69 *A. thaliana* accessions 23. For both PC genes and TE genes, our presence–absence variability is higher for gene models compared to gene loci, probably because our gene annotation pipeline misses lowly expressed genes in the four tissues we used. The PC gene model variability in the study from the Mercier lab 23 is substantially higher than in our data. This likely reflects the fact that while we used pan-genome coordinates to match genes between accessions, they used an orthogroup search approach. Consider this case: a gene is both present and has a gene model in every accession, but in three accessions, a shorter gene model was annotated. The orthogroup approach would identify

this case as two gene families residing in the locus and call both gene models dispensable, while our approach would treat this locus as one and identify both gene and gene model as core. **c**, Gene presence frequency distribution for TE genes. **d**, *A. lyrata* synteny and sequence similarity analysis for TE genes (TAIR10 and new) with the same 80% sequence-similarity stringency as in Fig. 6. TE genes are plotted by their gene IDs along the chromosomes. The small number of data points compared to PC genes is because most TE genes were not found in the *A. lyrata* genome. **e**, Functional domains ('NA' indicates genes with no match) for genes grouped by new vs. old and PC vs. TE. **f**, The same for PC genes grouped by frequency of presence. **g**, The same PC and TE genes grouped by ancestral status. **h**, The distribution of PC and TE genes along all five chromosomes grouped by frequency of presence. **i**, The same grouped by ancestral status.

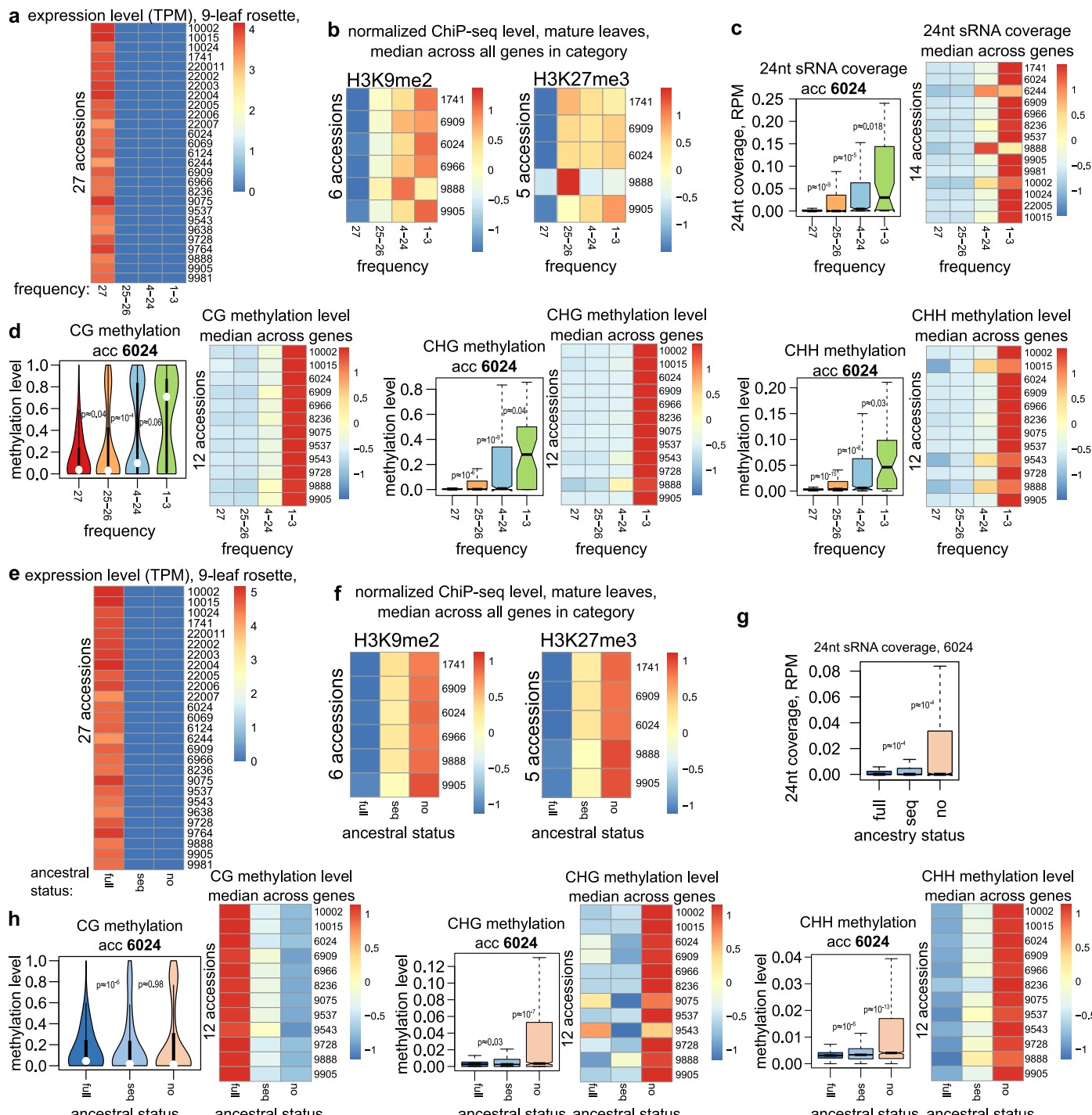

**Extended Data Fig. 6 | Gene silencing and expression by gene presence frequency and ancestral status. a**, Median expression vs. frequency for all accessions. **b**, H3K9me2 and H3K27me3 54 genes vs. their frequency. Values plotted are medians of quantile- and input-normalized ChIP-seq signals. **c**, 24-nt sRNAs coverage of genes of different frequencies. Left: distribution in flowers of accession 6024. Right: median coverage. **d**, Locus-wide CHG, CG and CHH methylation levels in leaves 53 for genes of different frequency. Left: distribution in accession 6024. Right: median levels. **e**, Expression (median TPM) vs. ancestral status for all accessions. **f**, H3K9me2 and H3K27me3 54 vs. ancestral status. Values plotted are medians of quantile- and input-normalized ChIP-seq

signals. **g**, 24-nt sRNAs locus-wide coverage in accession 6024, flowers, for genes of different ancestral status. **h**, Methylation levels in leaves 53 for genes of different ancestral status. Left: accession 6024. Right: median methylation levels. All heatmaps were created using the pheatmap package in *R*, and all, except for expression, are scaled by row to facilitate within- and between-accession comparison. Box plots show the median and the 25th and 75th percentiles (box bounds), and the smallest and largest values within 1.5× the interquartile range (whiskers); outliers are not plotted. Significance of median differences was assessed using two-tailed Wilcoxon tests. Sample sizes are shown above the top boxplot in Fig. 6i–j.

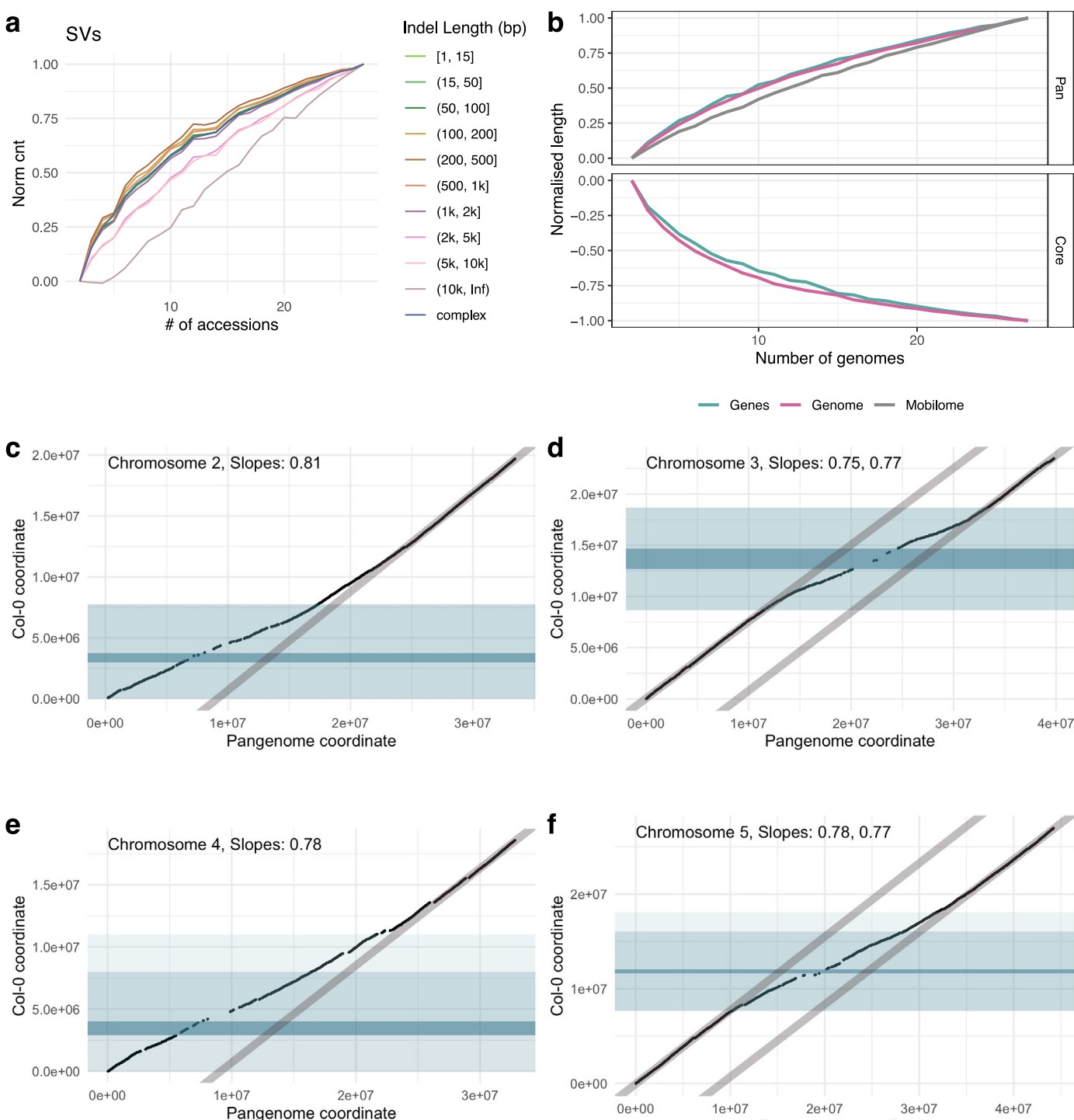

**Extended Data Fig. 7 | Extended pan-genome analysis. a**, Saturation curves for sSVs for different lengths. Each curve is drawn through points that are the averaged values from 20 repetitions. The curves were normalized to start and end at the same points. Larger sSVs saturate more quickly. **b**, Normalized saturation curves for different genome components: the dependence on sample size for the union ('pan') and intersection ('core') sequence length, separately for the full genome, the mobile-ome and the gene-ome. Within the 'pan' curves, the mobile-ome saturates the fastest, the gene-ome the slowest and the entire genome at an intermediate rate. Gene-ome and whole genome 'core' curves show similar trends. **c**–**f**, Pan-genome vs. reference genome coordinates for chromosomes 2–5. The pericentromeric region (light blue) shows a higher 'dilution' of the spatial coordinates due to the increased number of SVs in this region. The centromeric region is dark blue.

*Double-anonymous peer review submissions: write DAPR and your manuscript number here instead of author names.*

# Reporting Summary

## Statistics

For all statistical analyses, confirm that the following items are present in the figure legend, table legend, main text, or Methods section.

| n/a | Confirmed | |
|---|---|---|
| ☐ | ☒ | The exact sample size (*n*) for each experimental group/condition, given as a discrete number and unit of measurement |
| ☐ | ☒ | A statement on whether measurements were taken from distinct samples or whether the same sample was measured repeatedly |
| ☐ | ☒ | The statistical test(s) used AND whether they are one- or two-sided *Only common tests should be described solely by name; describe more complex techniques in the Methods section.* |
| ☐ | ☒ | A description of all covariates tested |
| ☐ | ☒ | A description of any assumptions or corrections, such as tests of normality and adjustment for multiple comparisons |
| ☐ | ☒ | A full description of the statistical parameters including central tendency (e.g. means) or other basic estimates (e.g. regression coefficient) AND variation (e.g. standard deviation) or associated estimates of uncertainty (e.g. confidence intervals) |
| ☐ | ☒ | For null hypothesis testing, the test statistic (e.g. *F*, *t*, *r*) with confidence intervals, effect sizes, degrees of freedom and *P* value noted *Give P values as exact values whenever suitable.* |
| ☒ | ☐ | For Bayesian analysis, information on the choice of priors and Markov chain Monte Carlo settings |
| ☒ | ☐ | For hierarchical and complex designs, identification of the appropriate level for tests and full reporting of outcomes |
| ☐ | ☒ | Estimates of effect sizes (e.g. Cohen's *d*, Pearson's *r*), indicating how they were calculated |

*Our web collection on statistics for biologists contains articles on many of the points above.*

## Software and code

Policy information about availability of computer code

| Data collection | No software was used. |
|---|---|
| Data analysis | Our code and scripts are available on GitHub: 1 https://github.com/Gregor-Mendel-Institute/1001Gplus_paper 2 https://github.com/iganna/pannagram 3 https://github.com/weigelworld/auto-asm The following software was used (not all have versions; all are in paper): Canu v1.71 Arrow v2.3.2 Pilon v1.22 Bionano Access v1.5 Bionano Solve v3.6 RagTag v1.1.1 pbmm2 v1.3.0 samtools v1.9 NucFreq v0.1 RepeatMasker v4.0.9 tRNAscan-SE v2.0.6 EDTA v1.9.7 Pannagram v2.0.0-beta MAFFT (ref 94) |

PGGB (Jan 2024)
VCF v1.54.0
NCBI BLAST server
Augustus v3.3.3
BUSCO v4.0.1
Liftoff v1.63
SNAP (ref 100)
Cufflinks (ref 101)
EvidenceModeler (ref 99)
blastp v 2.0.11
STAR v2.7.1
Subread v2.0.1
cutadapt v2.4
STAR (ref 104)
deeptools (ref 106)
bed-tools v.2.27.1
TrimGalore (https://github.com/FelixKrueger/TrimGalore)
Bismark (ref 109)
minimap2 v2.16
samtools v1.9
Jellyfish v2.3.0
findGSE (ref 116)
BLAST v2.2.29
UniProt DB (v. 2024_06).
Mummer4
BWA-MEM v0.7.17
Picard tools
GATK HaplotypeCaller v4.3
vcfwave
Nucmer
https://github.com/al2na/methylKit/issues/96

For manuscripts utilizing custom algorithms or software that are central to the research but not yet described in published literature, software must be made available to editors and reviewers. We strongly encourage code deposition in a community repository (e.g. GitHub). See the Nature Portfolio guidelines for submitting code & software for further information.

## Data

Policy information about availability of data

All manuscripts must include a data availability statement. This statement should provide the following information, where applicable:

- Accession codes, unique identifiers, or web links for publicly available datasets
- A description of any restrictions on data availability
- For clinical datasets or third party data, please ensure that the statement adheres to our policy

Raw sequencing data (PacBio CLR and Illumina PCR-free short reads) and genome assemblies have been deposited in the European Nucleotide Archive (https://www.ebi.ac.uk/ena/browser/home) under project accession number PRJEB73474 (ERP158243). Illumina PCR-free short reads for 61 additional accessions used to investigate the contribution of satellite repeats can be accessed under project accession number PRJEB73476 (ERP158245).

BS-seq data from mature leaves of 14-leaf rosettes are from the 1001 Genomes Project (Kawakatsu et al, 2016) and are available under GEO accession number GSE43857.
ChIP-, RNA-, and sRNA-seq data were likewise previously published (Kornienko et al, 2023). ChIP-seq data from 14-leaf rosettes are available under GEO accession number GSE226682; RNA-seq data from seedlings, 9-leaf rosettes, flowers, and pollen are available under GEO accession number GSE226691; and sRNA-seq data from flowers are available under the GEO accession number GSE224571.

Several widely used public databases were used in the analyses:
NCBI Protein Blast (https://www.ncbi.nlm.nih.gov/protein), UniProtKB} (version 2024\_06) (https://www.uniprot.org/), and
the TAIR10 genome annotation (\https://www.arabidopsis.org/).

## Research involving human participants, their data, or biological material

Policy information about studies with human participants or human data. See also policy information about sex, gender (identity/presentation), and sexual orientation and race, ethnicity and racism.

| Reporting on sex and gender | Work is on plants |
|---|---|
| Reporting on race, ethnicity, or other socially relevant groupings | Work is on plants |
| Population characteristics | Work is on plants |

| | |
|---|---|
| Recruitment | Work is on plants |
| Ethics oversight | Work is on plants |

Note that full information on the approval of the study protocol must also be provided in the manuscript.

# Field-specific reporting

Please select the one below that is the best fit for your research. If you are not sure, read the appropriate sections before making your selection.

☒ Life sciences ☐ Behavioural & social sciences ☐ Ecological, evolutionary & environmental sciences

For a reference copy of the document with all sections, see nature.com/documents/nr-reporting-summary-flat.pdf

# Life sciences study design

All studies must disclose on these points even when the disclosure is negative.

| | |
|---|---|
| Sample size | Sample was selected to cover global diversity; no conclusions depend on sample size. |
| Data exclusions | No data were excluded. |
| Replication | Please note that this study contains no experiments designed to test a specific hypothesis (like a drug trial). Replication was only used for a few types of measurements (e.g. the previously published ChIP-seq data) and the sample sizes used are given in the relevant figures. |
| Randomization | There are no experimental groups to randomize in this study. |
| Blinding | There are no experimental groups to be blind to in this study. |

# Reporting for specific materials, systems and methods

We require information from authors about some types of materials, experimental systems and methods used in many studies. Here, indicate whether each material, system or method listed is relevant to your study. If you are not sure if a list item applies to your research, read the appropriate section before selecting a response.

## Materials & experimental systems

| n/a | Involved in the study |
|---|---|
| ☒ | ☐ Antibodies |
| ☒ | ☐ Eukaryotic cell lines |
| ☒ | ☐ Palaeontology and archaeology |
| ☒ | ☐ Animals and other organisms |
| ☒ | ☐ Clinical data |
| ☒ | ☐ Dual use research of concern |
| ☐ | ☒ Plants |

## Methods

| n/a | Involved in the study |
|---|---|
| ☒ | ☐ ChIP-seq |
| ☒ | ☐ Flow cytometry |
| ☒ | ☐ MRI-based neuroimaging |

## Dual use research of concern

Policy information about dual use research of concern

### Hazards

Could the accidental, deliberate or reckless misuse of agents or technologies generated in the work, or the application of information presented in the manuscript, pose a threat to:

| No | Yes | |
|---|---|---|
| ☒ | ☐ | Public health |
| ☒ | ☐ | National security |
| ☒ | ☐ | Crops and/or livestock |
| ☒ | ☐ | Ecosystems |
| ☒ | ☐ | Any other significant area |

## Experiments of concern

Does the work involve any of these experiments of concern:

| No | Yes | |
|----|-----|---|
| ☒ | ☐ | Demonstrate how to render a vaccine ineffective |
| ☒ | ☐ | Confer resistance to therapeutically useful antibiotics or antiviral agents |
| ☒ | ☐ | Enhance the virulence of a pathogen or render a nonpathogen virulent |
| ☒ | ☐ | Increase transmissibility of a pathogen |
| ☒ | ☐ | Alter the host range of a pathogen |
| ☒ | ☐ | Enable evasion of diagnostic/detection modalities |
| ☒ | ☐ | Enable the weaponization of a biological agent or toxin |
| ☒ | ☐ | Any other potentially harmful combination of experiments and agents |

# Plants

| Seed stocks | No new seed stocks were collected: all are publicly avaiable. |
|----|----|
| Novel plant genotypes | None. |
| Authentication | DNA and RNA sequences were compared to public SNP data whenever possible. |

