## [Peer Review file · Nature Genetics]

A comparison of 27 *Arabidopsis thaliana* genomes and the path toward an unbiased characterization of genetic polymorphism

Corresponding Author: Dr Magnus Nordborg

Version 0:

Decision Letter:

30th Jul 2024

Dear Dr Nordborg,

Your Article, "Towards an unbiased characterization of genetic polymorphism" has now been seen by 3 referees. You will see from their comments copied below that while they find your work of considerable potential interest, they have raised quite substantial concerns that must be addressed. In light of these comments, we cannot accept the manuscript for publication, but would be very interested in considering a revised version that addresses these serious concerns.

We hope you will find the referees' comments useful as you decide how to proceed. If you wish to submit a substantially revised manuscript, please bear in mind that we will be reluctant to approach the referees again in the absence of major revisions.

To guide the scope of the revisions, the editors discuss the referee reports in detail within the team with a view to identifying key priorities that should be addressed in revision. In this case, we think all three referees have provided constructive reviews aimed at strengthening the analyses and improving the presentation. Importantly, Reviewer #1 notes that the title might need to be revised to more accurately represent the findings. We ask that you address all referee comments as thoroughly as possible with appropriate revisions. Please do not hesitate to get in touch if you would like to discuss these issues further.

If you choose to revise your manuscript taking into account all reviewer and editor comments, please highlight all changes in the manuscript text file. At this stage we will need you to upload a copy of the manuscript in MS Word .docx or similar editable format.

*2) If you have not done so already please begin to revise your manuscript so that it conforms to our Article format instructions, available here. Refer also to any guidelines provided in this letter.

*3) Include a revised version of any required Reporting Summary: <https://www.nature.com/documents/hr-reporting-summary.pdf>
It will be available to referees (and, potentially, statisticians) to aid in their evaluation if the manuscript goes back for peer review.
A revised checklist is essential for re-review of the paper.

Please be aware of our guidelines on

digital image standards.

Link Redacted

If you wish to submit a suitably revised manuscript we would hope to receive it within 6 months. If you cannot send it within this time, please let us know. We will be happy to consider your revision so long as nothing similar has been accepted for publication at Nature Genetics or published elsewhere. Should your manuscript be substantially delayed without notifying us in advance and your article is eventually published, the received date would be that of the revised, not the original, version.

Nature Genetics is committed to improving transparency in authorship. As part of our efforts in this direction, we are now requesting that all authors identified as 'corresponding author' on published papers create and link their Open Researcher and Contributor Identifier (ORCID) with their account on the Manuscript Tracking System (MTS), prior to acceptance. ORCID helps the scientific community achieve unambiguous attribution of all scholarly contributions. You can create and link your ORCID from the home page of the MTS by clicking on 'Modify my Springer Nature account'. For more information please visit please visit www.springernature.com/orcid.

Thank you for the opportunity to review your work.

Sincerely,
Wei

Wei Li, PhD
Senior Editor
Nature Genetics
www.nature.com/ng

Reviewers' Comments:

Reviewer #1:

Remarks to the Author:

Igolkina et al. assembled 27 de novo genomes of Arabidopsis and to explore genome evolution and estimate the true rate of diversity. Contrasting other plant species with much larger genomes, the Arabidopsis genome and genes are highly conserved, with 87% of genes being core across all accessions, and most variation due to transposable elements, centromere sequence, and TE associated genes. The authors also estimate nucleotide diversity in these accessions and compare this to findings from the 1001 project. A significant portion of true diversity was missed using short read data, which has important ramifications for the broader community. This work contributes significantly to our understanding of genetic diversity in Arabidopsis, marking the publication of the species' second pan-genome, with a previous one reported in Nature Genetics in April. I read this paper with interest and think it represents important findings for the field. I have a few concerns that should be addressed, and a few suggestions I would like to see incorporated to strengthen the analyses, but may be beyond the scope of this current manuscript.

1. I have some concerns about the genome assembly. The authors report 27 de novo reference genomes assembled using older PacBio sequencing technology, with subsequent polishing performed using Illumina short reads. This approach appears to contradict the manuscript's central argument: that short read-based estimates of nucleotide diversity, mutation load, and polymorphisms within a species are often underestimated due to the inherent ambiguities in mapping short reads. The reliance on short read data for genome polishing might undermine this claim, raising questions about the accuracy of the diversity estimates presented.

The authors also used TAIR10 as a reference for anchoring contigs and scaffolding, which might limit the detection of large-scale structural rearrangements.

All 27 assemblies are ~120 Mb in total length, despite 10-20% variation in genome size. Previous de novo Arabidopsis genomes have assemblies ranging from 128 to 148 Mb, with much better representation of centromeres, rRNAs, and other mostly non-coding sequence. The title of the paper suggests an unbiased characterization of genetic polymorphism across these genomes. However, this claim seems potentially misleading if significant portions of the genomes (probably megabases) remain unassembled. I would argue that outside of gene space, this study does not provide an unbiased characterization of polymorphisms as millions of bases are unassembled for each accession.

2. 69 Arabidopsis genomes were recently reported in Nature Genetics including 48 near complete genomes assembled with

PacBio HiFi data. This reference is mentioned in the paper, but the results from Lian et al. are not really incorporated here. Were any accessions in common between these two projects? If so, how do the genomes compare? It's hard to tell as Lian et al. 2024 used common names and Igolkina et al. used numbered accessions for Arabidopsis ecotypes. Are the gene spaces similar between both papers?

Ideally, it would be great to incorporate all of the genotypes, especially the wealth of very high-quality data generated by Lian et al. into the same pan-genome graph, but this may be beyond the scope of the present manuscript, and would delay release of these important resources.

3. The authors provide to my knowledge, the first estimation of true nucleotide diversity using de novo assembled genomes rather than short read based resequencing. The finding that 25-40% of diversity is missing from previous estimates is important and really interesting. How was within genome heterozygosity handled during assembly, pan genome graph construction, and downstream analyses? These accessions are presumably inbred, but some degree of polymorphism still exists, and the haploid genome assembly is only capturing one haplotype. Could this be a source of some of the SNP based polymorphisms that were missed using Illumina only approaches? SNPs were only identified for on average 84% of the genome from the 1001 genome project, how do SNPs compare in regions with high coverage or missing data in the Illumina resequencing and de novo assemblies? The HiFi data from Lian et al. is arguably higher quality than the data reported here, and would actually be ideal for testing the true rate of polymorphisms. Again, this analysis would be interesting, but perhaps beyond the scope of this manuscript.

Nucleotide diversity is reported in Figure 2C, but not really described. How does π compare to the 1001 project?

Somewhat related, I was surprised to see Pearson correlations of near 1 for gene expression data mapped to TAIR10 vs the newly assembled genomes. That's a good relief!

4. I have reservations about the introduction of the term 'geneome.' It is only one letter different from 'genome,' which could lead to confusion, and it is unlikely that 'geneome' will gain wide acceptance within the scientific community. The authors argue that the broad conservation of gene space in Arabidopsis renders the term 'pan-genome' inaccurate, especially in comparison to its original use in prokaryotes. However, the data presented do indicate significant variation within the Arabidopsis genome and even more so in other plant species, such as maize, where over half the gene space varies or is dispensable between genotypes. While the choice of terminology ultimately rests with the authors, I believe that 'geneome' may not resonate well or be readily adopted by the broader community, myself included.

In a similar vein, I am not sure the title accurately represents the paper or the findings. Only a few paragraphs of the results quantify missing polymorphisms from previous estimations, and this paper is mostly reporting a pan-genome of Arabidopsis, at least the way it is currently presented. Again, this is up to the author's discretion, but including 'Arabidopsis' in the title would be helpful, or change it to better reflect the findings.

Minor:

1. The authors state "Specifically, long presence alleles tend to be rare, while short ones tend to be common—suggesting that sSVs are mainly caused by long insertions and short deletions." I would guess many of the short SVs are microsatellites, and these could accumulate due to replication slippage, unequal crossing over or segmental duplications, and not insertions of TEs followed by purging (though of course, this also occurs frequently within Arabidopsis).

2. Is there a statistical test for this? "Segregating genes are somewhat enriched near centromeres, while fixed genes are clearly more common in the arms."

Reviewer #2:

Remarks to the Author:

In this manuscript, Igolkina et al. aim at providing an extensive and reference free characterization of genetic polymorphisms in the model plant *A. thaliana* by performing a thorough analysis of the genomes of 27 natural strains (representative of the genetic diversity of the species) that they de novo assemble using PacBio continuous long reads. To align and compare these whole-genome assemblies they develop a multiple-alignment pipeline (Pannagram), which they then use to detect and characterize structural variants within a pan-genome coordinate system. They find extensive structural variation, accounting for 40% of pan-genome size, in large parts contributed by transposable elements (TEs), which they analyze in a novel graph-based clustering approach to identify TE families, some of which appear to have not been previously identified in *A. thaliana*. In contrast, the genic portion is much more conserved with only a minor fraction, again appearing to be contributed by TEs, segregating across strains. Finally, they show that approaches based on short-read and a reference genome to identify SNPs or measure transcription levels or methylation states are strongly affected (between 25 to 45% of SNPs missed) by this unaccounted-for structural variation.

The tour de force attempted here by the authors pushes forward several important methodological advances that will be key to extract the most out of the complete genome sequences that are now being produced, notably those with higher level of TE activity and population structure than humans and where pan-genome graph tools perform poorly. Yet, the manuscript fails to present the myriad of analyses in a cohesive way and the conclusions reached are mostly not novel. The high level of TE activity in the *A. thaliana* genome was already well known, and it is no news either that TE annotations are far from perfect. The authors do provide one convincing evidence of a novel LTR retroelement potentially introduced by horizontal gene transfer from species outside Brassicaceae but only superficially describe it (Rech et al. Nat Comm 2022 use the TE characterization tool PASTEC for that purpose). Moreover, the conclusion that short-read based analysis of SNP and methylation polymorphisms were biased by cryptic structural variation has already been established by previous work from the same group (Jaegle et al. Genome Biology 2023). Finally, although the multiple-alignment methods developed by Igolkina et al. will be of high value to the community, the 27 pan-genome alignment produced here is likely of restricted use

by itself as the addition of a single new genome of interest requires the entire dataset to be realigned.

Major points:

- The authors equate the repertoire of repeated structural variation with a mobile-ome following the definition “the mobile-ome refers to the collection of insertions and deletions that are likely to have occurred recently and are therefore not fixed in our sample” (Methods, p2 last §). However, the mobile-ome is usually understood as “the set of TE families with transposition activity” (cf definition in Quadrana et al. eLife 2016). However, not all sequences in the authors’ repertoire of repeated SVs have the ability to mobilize and therefore are not mobile per se. Some may be the remnants of past mobilization events (e.g. solo-LTRs, truncated LINE insertions), while some may be repeated and structurally variable due to other mechanisms than mobilization (e.g. arrays of gene duplicates). If the authors want to use a looser definition that englobes sequences with no mobilization ability, they should state so clearly in the main text or perhaps coin another term (repetome would seem to fit better what the authors have in mind here).
- The clarity of the manuscript suffers from a major organization issue. Extended data figures are disorganized (Ext. Data Fig. 6 comes after 7 and 8), hard to understand (Ext. Data Fig. 8 is impossible to follow for non graph-alignment experts), missing labels (y-axis Ext. Data Fig. 7A), or with insufficient or unclear legends (supplementary note 3 see below)
- Although the practicality of focusing on the SVs detected by Pannagram rather than by PGGB is very clear, the bias that this choice introduces should be characterized at minima as PGGB identified almost twice as many variants which can be found across the whole genome and not only near centromeres (Extended Data Fig. 7B).

Minor comments:

The authors use TE families throughout the manuscript to refer to TE superfamilies (COPIA78 is a TE family, while COPIA is a superfamily).

Extended Data Fig. 7A: missing y-axis label

Page 4 §7: define “length polymorphisms”

Extended Data Fig. 6 comes after 7 and 8

Fig. 3C: what are seSVs?

the colors in supplementary note 3 should be changed to a more contrasted palette, as we cannot tell which superfamily is missing from the complete panel (only 7 distinct colors vs 8 in legend but the color of Copia, Gypsy, and LINEs are so close that we cannot tell which one)

Page 4: How are the deletions and insertions of Ext. Data Fig. 9 assigned is defined on page 5. Reorder

Fig. 4C-D: missing y-label

Page 5 §2: “with respect to their overlap” does not properly reflect the blast search for sequence homology that underlies this analysis, please rephrase

Page 7 §2: the example of putative new mobile element presented in Ext. Data Fig 17 is not exclusive to *A. thaliana* as blast clearly find hits in *A. arenosa* contradicting the claim that “they are exclusive to *A. thaliana*”

Page 7 §3: Given the conclusion that methylation levels from BS-seq are easily affected by structural variation, the presentation of results on methylation levels (Ext. Data Fig. 18) requires a minimal explanation of the method used to assess them.

Page 7 §3: “Methylation levels on sSVs without annotated TE content show the opposite pattern, which is expected given that the subset of these sSVs that are nevertheless part of the mobile-ome (and hence highly methylated), are mostly rare in the sampled genomes.” I understand that only a fraction of these sSVs are true TEs and therefore highly methylated, but not why this should result in decreasing levels of DNA methylation as allele frequency of the sSVs increase.

Ext. Data Fig. 19 is mentioned before the RNAseq it is based on

Fig. 6C: There seems to be a hotspot on *Iyrata* Chr 5 that contributes a large fraction of the new protein coding genes identified in *A. thaliana*. Could this be the result of a homology with TE sequence that was somehow missed in the reference genome? Indeed, new pc genes appear to be enriched in pericentromeres, lowly expressed and epigenetically silenced they are likely in large part corresponding to TE genes.

Reviewer #3:

Remarks to the Author:

The manuscript by Inolkina and colleagues report the development of a new tool (Pannagram) to analyse and visualise genetic polymorphisms (SNPs and SVs) in pangenomes. The authors produced and analysed a dataset of high quality de novo assemblies for 27 representative *Arabidopsis thaliana* accessions. For this, they used two methods: their newly developed Pannagram tool and PGGB. The authors characterize SVs in two compartments: the gene-ome (gene content) and mobile-ome (TE content). A comparison of SNP calling between short read (NGS) data and assemblies (WGA) is provided.

The originality of the work comes from the development of Pannagram. In contrast to pangenome graphs, Pannagram allows to anchor SVs in a reference genome in order to easily visualise them.

The data is of high quality and nicely presented however some figures are not visible (see below). The conclusions are well supported by the data. The fact that short read data does not allow neither the proper characterization of SVs nor an exhaustive SNP characterization is not new in the field. However here the authors quantify the SNPs identified in both cases (NGS vs WGA, Supp Fig 11).

Recent work in tomato or Brassica oleracea (Alonge et al, Cell, 2020; Li et al, Nature Genetics 2024, refs 5 and 21) have contributed to our understanding of the importance of SVs in phenotypic diversity. Here the work represents a computational tour de force both for data analysis and tool development. However, as written, the manuscript lacks details about the biology behind the newly discovered SVs. How can Pannagram increase the detection of causal SVs in SV-GWAS

analyses for instance seems out of the scope of the present work. Nevertheless a more detailed description of the main findings could be informative. For "mobile-ome" the newly discovered TE families could be more precisely described: were the insertions checked for the presence of target site duplications (TSDs) ? For "gene-ome" the function of genes associated with SVs is not clearly detailed. For example, Supp Note page 5: the authors mention that "new genes were strongly enriched for being present in 2 or more copies" but nothing is detailed concerning their function. Are they enriched for genes involved in interaction with the environment? In the Methods page 3 a BLAST search for "disease" and "receptor" terms is performed but this doesn't seem to be much elaborated in the text. Finally complex SVs are detected but not described in the text (or may I have missed this?). These SVs are especially important if the Pannagram tool is to be implemented in other, non autogamous, species for instance.

Other comments:

- I could not find the metrics of the dataset (coverage for PacBio CLR, read length, N50 for the Canu assemblies)
- Figure 2: I found chromosome 4 more striking to illustrate SVs.
- mobile-ome: "genetic elements that are actually mobile": these are TE polymorphisms. The term "actually" does not give any indication about the dynamics of mobility during evolution. TE polymorphisms can be more or less recent and still segregating.
- Why did the authors use TAIR10 versus Araport11?
- Page 12/Supp Fig 13 please precise in the text ALN=aligned, NOTAL=not aligned (only described in Supp Fig 12).
- Page 8: coverage information seems to be missing for the 27 datasets.
- Supp Fig 11 is one of the main outputs of the analysis (performance of SNP calling in short read data compared to whole genome assemblies), it could deserve appearing as main figure. Although the benefit of using proper reference genomes is not new, it is nicely quantified here.
- Supp Fig 5 I did not understand the merging solution (for instance red arrow in the bottom right panel).
- TE families in the text correspond to TE "Superfamilies". LTR Copia is a superfamily while COPIA78 is a family for instance.
- Supp Note page 3: about LTR Copia, consistent with their being active (reference missing here).
- Supp Fig 4: color code missing in B (same as C).
- Some figures are not visible: Supp Fig 3
- Abstract: "in contrast to this the genic portion is highly conserved" (it would be useful to precise) in this autogamous species.
- Page 7 typo They
- Fig 6 I/J: are these genes enriched for TE ORFs?
- Page 10: mapping algorithms contribute to discrepancies: are there any examples?
- Page 11: "We found thousands of examples of complete TEs". Are they supported by TSDs?
- Page 12: "These methods work well in humans". This is questionable as human genomics is centered on genes, please provide a reference.
- Last sentence about the usefulness of graphs in complex genomes seems controversial, especially given that the authors are not apparently maize genomicists. I wonder if a reference could be provided.

Version 1:

Decision Letter:

Our ref: NG-A65690R

25th Apr 2025

Dear Dr. Nordborg,

Thank you for submitting your revised manuscript "Towards an unbiased characterization of genetic polymorphism: a comparison of 27 *A. thaliana* genomes" (NG-A65690R). It has now been seen by the original referees and their comments are below. The reviewers find that the paper has improved in revision, and therefore we'll be happy in principle to publish it in Nature Genetics, pending minor revisions to satisfy the referees' final requests and to comply with our editorial and formatting guidelines.

Sincerely,
Wei

Wei Li, PhD
Senior Editor
Nature Genetics
www.nature.com/ng

Reviewer #2 (Remarks to the Author):

In this revised manuscript, Igolkina et al. have successfully addressed all the major technical and organizational issues raised during the first round of reviews, with only a few minor points left (cf below). Nonetheless, the novelty of this study still resides more in the approach and concepts than in the conclusions. Indeed, the methods that the authors develop here to analyze whole-genome assemblies through different angles are very much state-of-the-art and will represent a key resource for the community working on whole genome analyses in plants and beyond. In contrast, the conclusions, although very solid, mostly confirm or complete previous findings while the few novel results (previously undescribed TEs, newly identified genes) are just touched upon (which is entirely understandable as their in-depth investigation would represent full studies on their own). The whole-genome alignment generated here is also of little community use as it needs to be recalculated each time a new genome of interest is included, a drawback the authors readily acknowledged in their responses.

In this light, and in line with reviewer #1's comments, changing the title to focus more on the conceptual and methodological aspects of the study (that are again state-of-the-art) would help to better reflect the content of the manuscript.

Minor points:

P4, left column, last paragraph: The term "length variants" is cryptic, please define.

P4, right column, 2nd paragraph: the use of 'frequent' in the sentence "both types are more frequent in intergenic than in genic regions" is confusing as it could be understood as "more frequent in the population" but it seems here to refer instead to the number of sSVs that are intergenic rather than genic

P5, left column, 2nd paragraph, l3: replace 'more likely correspond' by 'more often correspond'?

P5, left column, 2nd paragraph, final sentence: not sure whether examination of flanking regions can truly help differentiate between decay rather than excision. What if excising TEs have a tendency to insert within TEs?

P5, left column, 3rd paragraph, final sentence: correct typo 'from perfectly' to 'from being perfectly'

P7, left column, 4th paragraph: Please mention how methylation data is analyzed (BSseq reads remapped on each genome assembly, if I understood correctly)

P7, left column, 4th paragraph: 'sSVs containing TEs or TE fragments are more variable' in what sense? In terms of DNA methylation levels?

P7, left column, 4th paragraph: 'consistent with a subset of these sSVs corresponding to un- or mis-annotated TEs' which subset? The ones that are highly methylated?

P7, left column, 5th paragraph: the claim that 'sSVs without annotated TE content behave similarly to sSVs containing TEs or TE fragments' is not at all what one extract from Extended Data Fig. 23. However there is indeed a subset of those that appear to behave similarly in terms of being highly methylated. Please correct accordingly.

P7, right column, 1st paragraph: Idem, with the claim to un-annotated sSVs that are part of the graph of nestedness are 'almost as highly methylated as previously annotated TEs'.

P7, right column, last paragraph: in the first version of this manuscript there was a hotspot in the *lyrata* genome of PC_new genes in 6C but they are gone now. Why? What did they correspond to? What were the changes responsible for their disappearance?

Reviewer #3 (Remarks to the Author):

Thanks for providing a clear and precise answer to my previous comments, in a tone that sounds very human and not AI based. Very much appreciated.

Although much is yet to be described in terms of SVs, the paper represents a landmark.

I could not systematically check all the fixed bugs in the text but I trust the authors.
Congratulations to the authors for a great job. Looking forward to all follow-up papers.

Response to reviewers

We thank the reviewers for helping us improve this paper. In addition to addressing your comments, we have thoroughly reworked the entire paper. In particular, the Pannagram pipeline was greatly improved, as described in the attached documents. This resulted in a whole new polymorphism and gene annotation. None of our conclusions were affected, but almost all figures and numbers had to be revised.

Reviewer #1

Remarks to the Author

Igolkina et al. assembled 27 de novo genomes of Arabidopsis and to explore genome evolution and estimate the true rate of diversity. Contrasting other plant species with much larger genomes, the Arabidopsis genome and genes are highly conserved, with 87% of genes being core across all accessions, and most variation due to transposable elements, centromere sequence, and TE associated genes. The authors also estimate nucleotide diversity in these accessions and compare this to findings from the 1001 project. A significant portion of true diversity was missed using short read data, which has important ramifications for the broader community. This work contributes significantly to our understanding of genetic diversity in Arabidopsis, marking the publication of the species' second pan-genome, with a previous one reported in Nature Genetics in April. I read this paper with interest and think it represents important findings for the field. I have a few concerns that should be addressed, and a few suggestions I would like to see incorporated to strengthen the analyses, but may be beyond the scope of this current manuscript.

Thank you!

1. I have some concerns about the genome assembly. The authors report 27 de novo reference genomes assembled using older PacBio sequencing technology, with subsequent polishing performed using Illumina short reads. This approach appears to contradict the manuscript's central argument: that short read-based estimates of nucleotide diversity, mutation load, and polymorphisms within a species are often underestimated due to the inherent ambiguities in mapping short reads. The reliance on short read data for genome polishing might undermine this claim, raising questions about the accuracy of the diversity estimates presented.

There is no contradiction here. Our argument about short-read mapping concerns problems that arise when reads are mapped to the "wrong" genome (see, e.g., Figure 8). In contrast, mapping short reads to the "right" genome, as is done during genome polishing, amounts to having multiple, more or less independent observations of the same thing, and is standard practice for improving sequencing quality (Rhie et al., *Nature*, 2021). Moreover, we have shown that CLR plus polishing with PCR-free reads results in consensus quality scores (QV) that are competitive with 40x HiFi alone (Rabanal et al., *Nucleic Acids Res*, 2022).

Of course our results only apply to the assembled regions of the genome, but this is clearly stated.

The authors also used TAIR10 as a reference for anchoring contigs and scaffolding, which might limit the detection of large-scale structural rearrangements.

We guarded against this by using BioNano maps for eight genomes, plus we used a version of TAIR10 carefully masked for satellite repeats and organellar insertions (Rabanal *et al.*, *Nucleic Acids Res*, 2022), which can lead to erroneous scaffolding. This is mentioned in the paper, and is an improvement over both Lian *et al.* (*Nature Genetics*, 2024) and Kang *et al.* (*Nature Commun*, 2023) which only used TAIR10 and hence had several scaffolding errors, as we deduced from inspection of their assemblies (our unpublished results).

All 27 assemblies are ~120 Mb in total length, despite 10-20% variation in genome size. Previous de novo *Arabidopsis* genomes have assemblies ranging from 128 to 148 Mb, with much better representation of centromeres, rRNAs, and other mostly non-coding sequence.

This is an inevitable consequence of the technology we used, as explained in the paper. It does not affect our conclusions, which are explicitly based only on the ~120 Mb shown (Figs. 1 and 2). Lian *et al.* (*Nature Genetics* 2024) were indeed able to capture more of the highly repetitive genome sequences, although their assemblies also lacked up to 15 Mb of sequence (Lian *et al.* Fig. 2a). More importantly, both Lian *et al.* (*Nature Genetics*, 2024) and we have a good grasp of what is missing in our genomes. We show that the missing sequence in our genomes is mostly centromere and rDNA sequence (Fig. 1), while TEs and other repeats in unplaced contigs add up to only ~0.5 Mb on average (Extended Data Figure 3).

The title of the paper suggests an unbiased characterization of genetic polymorphism across these genomes. However, this claim seems potentially misleading if significant portions of the genomes (probably megabases) remain unassembled. I would argue that outside of gene space, this study does not provide an unbiased characterization of polymorphisms as millions of bases are unassembled for each accession.

Our title is clearly aspirational: “**Towards** an unbiased...”. It is in a similar spirit to Rhie *et al.* (*Nature* 2021) “Towards complete and error-free genome assemblies of all vertebrate species”. We believe we are very transparent in terms of what we have and what we have NOT achieved with our efforts, but we are open to suggestions on how to make this even clearer. We are consulting with the editor whether a title such as “Towards an unbiased characterization of genetic polymorphism: a comparison of 27 *Arabidopsis thaliana* genomes” would work for them.

We agree that missing some portion of each genome could introduce biases in genome comparisons, but we make no claims about these regions. Furthermore, the regions missing in our study, i.e., rDNA arrays and centromeres, are affected by different mutational dynamics than the rest of the genome (see previous work from the Nordborg lab as well as the recent Henderson/Weigel collaboration paper Wlodzimierz *et al.* *Nature* 2023). There are relatively few transposable elements in these regions of the genome (11 to 74 per genome according to Wlodzimierz *et al.* *Nature* 2023), which is relevant to our mobile-ome section.

We clearly state that we make claims only for chromosome arms, and we note that Lian *et al.* (*Nature Genetics* 2024) also characterized only chromosome arms. This includes gene space *and* intergenic space, but not centromeres, telomeres, and rDNA clusters. Thus, we go considerably beyond “gene space”.

2. 69 Arabidopsis genomes were recently reported in Nature Genetics including 48 near complete genomes assembled with PacBio HiFi data. This reference is mentioned in the paper, but the results from Lian *et al.* are not really incorporated here. Were any accessions in common between these two projects? If so, how do the genomes compare? It's hard to tell as Lian *et al.* 2024 used common names and Iolkina *et al.* used numbered accessions for Arabidopsis ecotypes. Are the gene spaces similar between both papers?

Exact comparison of gene spaces will require common annotation of all the genomes; this is something we are working on. We have assembled a collection of ~600 genomes from various published (including 32 from Kang *et al.*, 66 from Wlodzimierz *et al.*, and 72 from Lian *et al.*) and unpublished sources (mostly HiFi, some CLR, some ONT), and a joint analysis of this will be the subject of a future paper. Nevertheless, the overall conclusions in Lian *et al.* (*Nature Genetics*, 2024) regarding universal and private genes and the classes in between are broadly similar to ours. Detailed differences are almost certainly due to differences in annotation and methods for clustering gene models and comparing these across genomes as well as the number and diversity of genomes considered (see our Extended Data Figure 20B). We could of course enumerate the precise numbers in Lian *et al.* and Kang *et al.* compared to ours at every step, but due to the above-mentioned technical differences, this is like comparing apples and oranges and not very useful. That said, in a few cases where we thought it was useful, we make explicit comparisons.

Ideally, it would be great to incorporate all of the genotypes, especially the wealth of very high-quality data generated by Lian *et al.* into the same pan-genome graph, but this may be beyond the scope of the present manuscript, and would delay release of these important resources.

Yes, this is beyond the scope of the present manuscript. As described above, we are working on an integrated analysis of all available *A. thaliana* long-read genomes. Combining hundreds of genomes of variable quality is a massive undertaking, but it is essential to enable a uniform annotation for meaningful downstream analysis. The methods and insights developed in this paper are the basis for this next step.

As an aside, note that the Lian *et al.* (*Nature Genetics*, 2024) genomes varied considerable in quality. First, 24 genomes were produced by older ONT technology. Second, the coverage for several HiFi genomes was relatively low, and Lian *et al.* had to use short read polishing, as we had to.

3. The authors provide to my knowledge, the first estimation of true nucleotide diversity using de novo assembled genomes rather than short read based resequencing. The finding that 25-40% of diversity is missing from previous estimates is important and really interesting.

We agree!

How was within genome heterozygosity handled during assembly, pan genome graph construction, and downstream analyses? These accessions are presumably inbred, but some degree of polymorphism still exists, and the haploid genome assembly is only capturing one haplotype.

In naturally inbreeding species such as *A. thaliana*, heterozygosity is generated by outcrossing, but it decays very rapidly during subsequent selfing. Residual heterozygosity is still common in lines collected from nature, easily seen as long tracts of heterozygosity. We routinely check this by mapping the original long reads to the respective assemblies. In the collection of ~600 long-read genomes mentioned above, including all published long-read assemblies for *A. thaliana*, we have discovered several genomes that are indeed from strains that were not completely inbred and that have substantial amounts of heterozygosity (including the three mentioned by Lian *et al.* in their paper). In 26 of our 27 genomes, we detect no heterozygosity at all. Accession 35-1 (ecotype ID: 22002) has some small regions in chromosome 3 with an increased read depth for the secondary allele, but this does not impact the quality of the final scaffolded assembly, since both the length of the final nuclear scaffolded assembly (Extended Data Figure 3) and the BUSCO scores are comparable to all other 26 assemblies (Complete: 99.2% [Single: 98.1%, Duplicated: 1.1%], Fragmented: 0.2%, Missing: 0.6%, n = 1,614). This is noted in Methods.

Could this be a source of some of the SNP based polymorphisms that were missed using Illumina only approaches?

This is very unlikely. There is only very limited residual heterozygosity in one of our lines (see previous comment). As we explain, the most common source is not being able to perform unambiguous mapping to the chosen reference genome (either the region is missing, or non-unique). These are regions that are often difficult to align even with full genome assemblies, and we discuss the problem of deciding what is actually a SNP (a difference due to a single nucleotide substitution) versus an alignment artifact. The relevant section has been clarified in the revision.

SNPs were only identified for on average 84% of the genome from the 1001 genome project, how do SNPs compare in regions with high coverage or missing data in the Illumina resequencing and de novo assemblies?

Please see Methods and Supplementary Note 5—errors are almost entirely due to read-mapping and alignment issues. We have also moved a supplementary figure to Fig. 8A to emphasize this.

The HiFi data from Lian *et al.* is arguably higher quality than the data reported here, and would actually be ideal for testing the true rate of polymorphisms.

Again, this analysis would be interesting, but perhaps beyond the scope of this manuscript.

It is not obvious that the data from Lian *et al.* are of higher quality or more complete for the parts of the genome covered (see above), but, more importantly, technical errors make a trivial contribution

to the discrepancies seen (as we note in the paper). The real challenge lies in aligning genomes and interpreting what we see.

Nucleotide diversity is reported in Figure 2C, but not really described. How does P_i compare to the 1001 project?

This figure is meant to illustrate that the pattern of structural variation across the genome mirrors the previously described pattern of SNP variation. It is discussed in this context. The figure shows synonymous P_i , which is not strongly affected by the additional variation uncovered in this paper. As noted in "Missing polymorphism", the average pairwise difference between two accessions in 1001 Genomes was 440k SNPs—with full genomes we estimate 600-800k, however most of this affects non-genic regions, where calculation of P_i are complicated by the need of a denominator (how many bp are assessed?).

Somewhat related, I was surprised to see Pearson correlations of near 1 for gene expression data mapped to TAIR10 vs the newly assembled genomes. That's a good relief!

We agree (although note that the most variable genes are often expressed at very low levels, and thus make a smaller contribution to the genome-wide correlation.)

4. I have reservations about the introduction of the term 'geneome.' It is only one letter different from 'genome,' which could lead to confusion, and it is unlikely that 'geneome' will gain wide acceptance within the scientific community. The authors argue that the broad conservation of gene space in Arabidopsis renders the term 'pan-genome' inaccurate, especially in comparison to its original use in prokaryotes. However, the data presented do indicate significant variation within the Arabidopsis genome and even more so in other plant species, such as maize, where over half the gene space varies or is dispensable between genotypes. While the choice of terminology ultimately rests with the authors, I believe that 'geneome' may not resonate well or be readily adopted by the broader community, myself included.

We had absolutely no intention of introducing a term that would be widely adopted. On the contrary, we deliberately used a rather ugly term to guard against this possibility, because biology has quite enough terms and jargon already. We found the term "gene-ome" (which we define) useful in the context of our paper as a contrast to "mobile-ome" (another terrible term that we also define). The widely used "pan-genome" is already problematic as different authors use it to mean different things (as we discussed). In the revision, we have carefully gone through the paper to ensure that all non-standard terminology is defined on first usage.

In a similar vein, I am not sure the title accurately represents the paper or the findings. Only a few paragraphs of the results quantify missing polymorphisms from previous estimations, and this paper is mostly reporting a pan-genome of Arabidopsis, at least the way it is currently presented. Again, this is up to the author's discretion, but including 'Arabidopsis' in the title would be helpful, or change it to better reflect the findings.

We do not disagree, however, a title that captured all we do would be long indeed. How about what we suggested above: “Towards an unbiased characterization of genetic polymorphism: a comparison of 27 *Arabidopsis thaliana* genomes”?

Minor comments

1. The authors state “Specifically, long presence alleles tend to be rare, while short ones tend to be common—suggesting that sSVs are mainly caused by long insertions and short deletions.” I would guess many of the short SVs are microsatellites, and these could accumulate due to replication slippage, unequal crossing over or segmental duplications, and not insertions of TEs followed by purging (though of course, this also occurs frequently within *Arabidopsis*).

This agrees with our observations, although microsatellites also seem to be part of long SVs, especially complex ones. An unexpected observation is that microsatellites leading to SVs are enriched in TEs, particularly in certain families within the HELITRON, ARNOLD and VANDAL superfamilies (see Extended Figure 18 for an example of this). As we state in the discussion, we believe an analysis based on the Ancestral Recombination Graph will shed considerable light on the molecular mechanisms that cause SVs, and this is being planned. Until then we can only make anecdotal observations.

2. Is there a statistical test for this? “Segregating genes are somewhat enriched near centromeres, while fixed genes are clearly more common in the arms. “

Sure. The difference is highly significant using a chi-square test. This has been noted.

Reviewer #2

Remarks to the Author

In this manuscript, Igolkina et al. aim at providing an extensive and reference free characterization of genetic polymorphisms in the model plant *A. thaliana* by performing a thorough analysis of the genomes of 27 natural strains (representative of the genetic diversity of the species) that they de novo assemble using PacBio continuous long reads. To align and compare these whole-genome assemblies they develop a multiple-alignment pipeline (Pannagram), which they then use to detect and characterize structural variants within a pan-genome coordinate system. They find extensive structural variation, accounting for 40% of pan-genome size, in large parts contributed by transposable elements (TEs), which they analyze in a novel graph-based clustering approach to identify TE families, some of which appear to have not been previously identified in *A. thaliana*. In contrast, the genic portion is much more conserved with only a minor fraction, again appearing to be contributed by TEs, segregating across strains. Finally, they show that approaches based on short-read and a reference genome to identify SNPs or measure transcription levels or methylation states are strongly affected (between 25 to 45% of SNPs missed) by this unaccounted-for structural variation.

The tour de force attempted here by the authors pushes forward several important methodological advances that will be key to extract the most out of the complete genome sequences that are now being produced, notably those with higher level of TE activity and population structure than humans

and where pan-genome graph tools perform poorly. Yet, the manuscript fails to present the myriad of analyses in a cohesive way and the conclusions reached are mostly not novel.

We have tried to improve the narrative, but are open to suggestions for how to make the presentation more cohesive. Regarding the novelty of the conclusions: Yes, they are perhaps not unexpected, but most have not been formally demonstrated before. If we have missed references to papers that have demonstrated our main points before, we would of course be more than happy to include these.

The high level of TE activity in the *A. thaliana* genome was already well known, and it is no news either that TE annotations are far from perfect.

Indeed, and we do not claim otherwise, citing previous results. But our approaches for detecting new TE families that are neither high copy nor look like known TEs is novel. In addition, what we call the “mobile-ome” is not quite the same as the “TE-ome”.

The authors do provide one convincing evidence of a novel LTR retroelement potentially introduced by horizontal gene transfer from species outside Brassicaceae but only superficially describe it (Rech et al. Nat Comm 2022 use the TE characterization tool PASTEC for that purpose).

We have added this citation, which we had missed—thanks for pointing it out. As for novel elements, we give several examples, including of completely novel ones with no similarity to previously annotated TEs in *A. thaliana*. This section has been improved, and more details added. But this is only one of several sections in the paper, and a follow-up paper that provides in-depth description and improved TE annotation is in progress.

Moreover, the conclusion that short-read based analysis of SNP and methylation polymorphisms were biased by cryptic structural variation has already been established by previous work from the same group (Jaegle et al. Genome Biology 2023).

This paper (which of course we cite) addresses a small subset of the issues covered in this paper, and does not attempt to understand the precise cause of the discrepancy by looking at the nature of mismapping. Ironically, reviewers of Jaegle et al. (*Genome Biology*, 2023) asked about this, and we responded that it was beyond the scope of the paper, and would be covered in the next one. Which it was and is.

Finally, although the multiple-alignment methods developed by Igoikina et al. will be of high value to the community, the 27 pan-genome alignment produced here is likely of restricted use by itself as the addition of a single new genome of interest requires the entire dataset to be realigned.

This is a property of any unbiased whole-genome alignment (including pan-genome graphs). As noted in our response to Reviewer 1, we are currently applying Pannagram to about 600 assemblies, and we will continue to update the alignment periodically as soon as the genomes are available. This will cover the needs of most users. Meanwhile it is possible for colleagues to run our pipeline themselves, including their own genomes. And of course we and others are developing methods to

genotype lines with only short-read data by mapping them to the whole data set. This is a central goal for human genetics.

Major points

- The authors equate the repertoire of repeated structural variation with a mobile-ome following the definition “the mobile-ome refers to the collection of insertions and deletions that are likely to have occurred recently and are therefore not fixed in our sample” (Methods, p2 last §). However, the mobile-ome is usually understood as “the set of TE families with transposition activity” (cf definition in Quadrana et al. eLife 2016). However, not all sequences in the authors’ repertoire of repeated SVs have the ability to mobilize and therefore are not mobile per se. Some may be the remnants of past mobilization events (e.g. solo-LTRs, truncated LINE insertions), while some may be repeated and structurally variable due to other mechanisms than mobilization (e.g. arrays of gene duplicates). If the authors want to use a looser definition that englobes sequences with no mobilization ability, they should state so clearly in the main text or perhaps coin another term (repetome would seem to fit better what the authors have in mind here).

We agree that our usage differs from that of Quadrana et al. (eLife, 2016), and we have clarified our definition to avoid any ambiguity. In particular, our mobile-ome is not identical to the TE-ome.

- The clarity of the manuscript suffers from a major organization issue. Extended data figures are disorganized (Ext. Data Fig. 6 comes after 7 and 8), hard to understand (Ext. Data Fig. 8 is impossible to follow for non graph-alignment experts), missing labels (y-axis Ext. Data Fig. 7A), or with insufficient or unclear legends (supplementary note 3 see below)

Spot on—putting together a paper of this size and complexity is hard, and there was still a number of embarrassing mistakes in the submitted version (in Methods, Extended Data, and Supplement). We realize this is annoying, and apologize. In the revised version we have hopefully fixed almost all, if not all these issues. In particular, we have expanded and clarified the graph and TE examples.

- Although the practicality of focusing on the SVs detected by Pannagram rather than by PGGB is very clear, the bias that this choice introduces should be characterized at minima as PGGB identified almost twice as many variants which can be found across the whole genome and not only near centromeres (Extended Data Fig. 7B).

Yes, this should be commented on. We have completely reworked the discussion of the differences between these approaches (including adding several nice examples), including why a comparison is hard. We identify several reasons why graph SVs tend to cover greater sequence length than Pannagram SVs, and we argue that “bias” is not the right word given that it is not clear what we are trying to estimate.

Minor comments

The authors use TE families throughout the manuscript to refer to TE superfamilies (COPIA78 is a TE family, while COPIA is a superfamily).

Indeed, there was some inconsistency in the terminology. We have reviewed all instances and strictly followed the hierarchical classification: class → superfamily → family.

Extended Data Fig. 7A: missing y-axis label

Figure has been replaced.

Page 4 §7: define “length polymorphisms”

Paragraph has been rewritten, clarifying this, *inter alia*.

Extended Data Fig. 6 comes after 7 and 8

Done.

Fig. 3C: what are seSVs?

It was a typo for sSVs, and we have corrected it.

the colors in supplementary note 3 should be changed to a more contrasted palette, as we cannot tell which superfamily is missing from the complete panel (only 7 distinct colors vs 8 in legend but the color of Copia, Gypsy, and LINEs are so close that we cannot tell which one)

We have updated the color scheme to a more contrasted one to enhance differentiation and maintain consistency with other figures on TE super-families.

Page 4: How are the deletions and insertions of Ext. Data Fig. 9 assigned is defined on page 5.

Definition has been added to caption.

Fig. 4C-D: missing y-label

Figure D is the histogram in Figure C, but normalized by the number of TE-related sSVs. We agree that this was obscure. Figure has been updated..

Page 5 §2: “with respect to their overlap” does not properly reflect the blast search for sequence homology that underlies this analysis, please rephrase

We agree and have rephrased the sentence..

Page 7 §2: the example of putative new mobile element presented in Ext. Data Fig 17 is not exclusive to *A. thaliana* as blast clearly find hits in *A. arenosa* contradicting the claim that “they are exclusive to *A. thaliana*”

Indeed, thanks for spotting this: all mobile element examples have been reworked. For the record, we also have examples that appear to be unique.

Page 7 §3: Given the conclusion that methylation levels from BS-seq are easily affected by structural variation, the presentation of results on methylation levels (Ext. Data Fig. 18) requires a minimal explanation of the method used to assess them.

Yes, this is an important point and we agree with the need for additional explanation. The methylation estimates bias we report in Fig. 8 refers to using the TAIR10 reference genome for estimating methylation levels in all non-reference accessions. It is when we use this genome we miscalculate methylation on structurally variable genes (and elements), and we compare the results to what we get when we map reads to their own genome. The methylation estimates for SVs in Extended Data Figure 18 (Extended Data Figure 23 in the revised manuscript) were obtained by mapping reads to their own genome so the problem described in Fig. 8 does not apply. We added a better explanation to the legend of the Extended Data Figure and the Methods.

Page 7 §3: "Methylation levels on sSVs without annotated TE content show the opposite pattern, which is expected given that the subset of these sSVs that are nevertheless part of the mobile-ome (and hence highly methylated), are mostly rare in the sampled genomes." I understand that only a fraction of these sSVs are true TEs and therefore highly methylated, but not why this should result in decreasing levels of DNA methylation as allele frequency of the sSVs increase.

We rewrote this passage to improve clarity.

Ext. Data Fig. 19 is mentioned before the RNAseq it is based on

Fixed.

Fig. 6C: There seems to be a hotspot on *Iyrata* Chr 5 that contributes a large fraction of the new protein coding genes identified in *A. thaliana*. Could this be the result of a homology with TE sequence that was somehow missed in the reference genome? Indeed, new pc genes appear to be enriched in pericentromeres, lowly expressed and epigenetically silenced they are likely in large part corresponding to TE genes.

In the new annotation and ancestry analysis, this hotspot no longer exists. We have also added an enrichment analysis that addresses the likely contribution of TE genes.

Reviewer #3

Remarks to the Author

The manuscript by Inolkina and colleagues report the development of a new tool (Pannagram) to analyse and visualise genetic polymorphisms (SNPs and SVs) in pangenomes. The authors produced and analysed a dataset of high quality de novo assemblies for 27 representative

Arabidopsis thaliana accessions. For this, they used two methods: their newly developed Pannagram tool and PGGB. The authors characterize SVs in two compartments: the gene-ome (gene content) and mobile-ome (TE content). A comparison of SNP calling between short read (NGS) data and assemblies (WGA) is provided.

The originality of the work comes from the development of Pannagram. In contrast to pangenome graphs, Pannagram allows to anchor SVs in a reference genome in order to easily visualise them.

The data is of high quality and nicely presented however some figures are not visible (see below). The conclusions are well supported by the data. The fact that short read data does not allow neither the proper characterization of SVs nor an exhaustive SNP characterization is not new in the field. However here the authors quantify the SNPs identified in both cases (NGS vs WGA, Supp Fig 11).

Recent work in tomato or *Brassica oleracea* (Alonge et al, Cell, 2020; Li et al, Nature Genetics 2024, refs 5 and 21) have contributed to our understanding of the importance of SVs in phenotypic diversity. Here the work represents a computational tour de force both for data analysis and tool development. However, as written, the manuscript lacks details about the biology behind the newly discovered SVs. How can Pannagram increase the detection of causal SVs in SV-GWAS analyses for instance seems out of the scope of the present work.

This is indeed out of scope, and will have to await a larger sample size (which is forthcoming, as noted in our response to Reviewer 1). We could have genotyped SVs found in the 27 genomes using short reads available for the full 1001 Genomes set, but it seems much more sensible to wait until we have better data. Our paper has a different focus than other papers that prioritized morphological and metabolic phenotypes, but ignored technical issues. (We note that we did look at methylation and expression phenotypes as a function of population frequency of genes.)

Nevertheless a more detailed description of the main findings could be informative. For "mobile-ome" the newly discovered TE families could be more precisely described: were the insertions checked for the presence of target site duplications (TSDs) ?

We certainly agree with the general point that these findings deserve to be described better, however, please note that a careful description over 200 putative new elements is a substantial undertaking that surely deserves a separate paper. Following your suggestion, we have started developing a pipeline to identify TSDs for our sSVs, and indeed we observe them, although the pattern is complex, presumably reflecting poorly understood target-site preferences. We have also started looking systematically for LTRs, and again we find interesting patterns (often confirming that elements containing TE annotation are simply mis-annotated). We are in close contact with people responsible for the TAIR12 annotation, and hope to contribute to an improved TE annotation in the near future. However, we are not ready to present these results in the current paper, and have instead opted to provide examples of what we find (including examples of new families with TSDs).

For "gene-ome" the function of genes associated with SVs is not clearly detailed. For example, Supp Note page 5: the authors mention that "new genes were strongly enriched for being present in 2 or

more copies" but nothing is detailed concerning their function. Are they enriched for genes involved in interaction with the environment? In the Methods page 3 a BLAST search for "disease" and "receptor" terms is performed but this doesn't seem to be much elaborated in the text.

This analysis has been greatly extended (see Fig. 6), and indeed your suspicion is correct: new genes look less like housekeeping genes and more like defense genes. Interestingly, new TEs also look more like TEs and less like genes, confirming that segregating TEs are indeed more likely to actually be TEs!

Finally complex SVs are detected but not described in the text (or may I have missed this?). These SVs are especially important if the Pannagram tool is to be implemented in other, non autogamous, species for instance.

We have reworked and expanded the comparison of the PGGB graph and Pannagram and in particular added Extended Data Figures 9-12 describing how complex regions and variants are handled. As for the nature and origin of these, that is the topic of several follow-up papers! We do not think there is a fundamental difference between selfing and outcrossing organisms in this respect, and Pannagram has already been run on several outcrossing species.

Other comments

- I could not find the metrics of the dataset (coverage for PacBio CLRs, read length, N50 for the Canu assemblies)

This information can be found in Supplementary Table 1.

- Figure 2: I found chromosome 4 more striking to illustrate SVs.

We agree that it's a better illustration of large-scale SVs, but it is a poorer illustration of their general absence, and the fact that this hides large amounts of smaller SVs.

- mobile-ome: "genetic elements that are actually mobile": these are TE polymorphisms. The term "actually" does not give any indication about the dynamics of mobility during evolution. TE polymorphisms can be more or less recent and still segregating.

Agreed, the term was meant to distinguish elements with segregating presence/absence polymorphisms (which must have been active recently, certainly within 1 Myr) from repetitive elements that are not segregating but still litter the genome (and could be much older). We have clarified this in the revision.

- Why did the authors use TAIR10 versus Araport11?

For historical reasons, mostly. In our next paper we will hopefully be able to use the forthcoming TAIR12 annotation.

- Page12/Supp Fig 13 please precise in the text ALN=aligned, NOTAL=not aligned (only described in Supp Fig12).

Fixed. We've also expanded the description in the supplemental notes to explain these two terms: "delta" file produced by Nucmer was used to determine the ALN (Aligned, marks blocks of aligned sequences between two genomes) and NOTAL (Not Aligned, highlights sections of a genome that didn't align with the other) regions.

- Page 8: coverage information seems to be missing for the 27 datasets.

This is either in Supplemental Tables (or published, e.g., for methylation data).

- Supp Fig 11 is one of the main outputs of the analysis (performance of SNP calling in short read data compared to whole genome assemblies), it could deserve appearing as main figure. Although the benefit of using proper reference genomes is not new, it is nicely quantified here.

Thanks, we agree, and have moved this supplementary figure to Fig. 8. It also makes the point that the main issue is alignment, and that defining errors is more subtle than most people are aware.

- Supp Fig 5 I did not understand the merging solution (for instance red arrow in the bottom right panel).

Yes, this was obscure: thanks for pointing it out. As noted in our response to Reviewer 2, much in Supplements could have been presented better. We have revised this caption as part of revision the whole section.

- TE families in the text correspond to TE "Superfamilies". LTR Copia is a superfamily while COPIA78 is a family for instance.

Yup, this has been fixed.

- Supp Note page 3: about LTR Copia, consistent with their being active (reference missing here).

Fixed.

- Supp Fig 4: color code missing in B (same as C).

Fixed.

- Some figures are not visible: Supp Fig 3

Fixed.

- Abstract: "in contrast to this the genic portion is highly conserved " (it would be useful to precise) in this autogamous species.

Fixed.

- Page 7 typo Theey

Fixed.

- Fig 6 I/J: are these genes enriched for TE ORFs?

This figure has been redone, and analyses added that indeed suggest this.

- Page 10: mapping algorithms contribute to discrepancies: are there any examples?

Yes! We've added a reference to a supplementary figure that provides two examples of how the same sequence pair can result in different alignments depending on the algorithm, illustrating this source of variation.

- Page 11: "We found thousands of examples of complete TEs". Are they supported by TSDs?

Indeed many are, and we also find LTRs, etc. However, as we noted above, we are simply not ready to release an improved TE annotation based on these results, and are working with the TAIR12 team on this. In this paper we only provide examples to give a flavor of what we find (of course we also release the full data should someone wish to annotate them!)

- Page 12: "These methods work well in humans". This is questionable as human genomics is centered on genes, please provide a reference.

Several papers (e.g. Evan Eichler's work) have looked at non-coding polymorphisms in humans, but we deleted the sentence in revision.

- Last sentence about the usefulness of graphs in complex genomes seems controversial, especially given that the authors are not apparently maize genomicists. I wonder if a reference could be provided.

Provided, and the authors of that reference told us that they agree with our conclusions.

Response to reviewers

Reviewer #2 (Remarks to the Author)

In this revised manuscript, Igolkina et al. have successfully addressed all the major technical and organizational issues raised during the first round of reviews, with only a few minor points left (cf below).

Nonetheless, the novelty of this study still resides more in the approach and concepts than in the conclusions. Indeed, the methods that the authors develop here to analyze whole-genome assemblies through different angles are very much state-of-the-art and will represent a key resource for the community working on whole genome analyses in plants and beyond. In contrast, the conclusions, although very solid, mostly confirm or complete previous findings while the few novel results (previously undescribed TEs, newly identified genes) are just touched upon (which is entirely understandable as their in-depth investigation would represent full studies on their own). The whole-genome alignment generated here is also of little community use as it needs to be recalculated each time a new genome of interest is included, a drawback the authors readily acknowledged in their responses.

In this light, and in line with reviewer #1's comments, changing the title to focus more on the conceptual and methodological aspects of the study (that are again state-of-the-art) would help to better reflect the content of the manuscript.

We have changed the title to better reflect the content of the manuscript.

Minor points:

P4, left column, last paragraph: The term "length variants" is cryptic, please define.

We define SVs in the previous section.

P4, right column, 2nd paragraph: the use of 'frequent' in the sentence "both types are more frequent in intergenic than in genic regions" is confusing as it could be understood as "more frequent in the population" but it seems here to refer instead to the number of sSVs that are intergenic rather than genic

Clarified.

P5, left column, 2nd paragraph, l3: replace 'more likely correspond' by 'more often correspond'?

Fixed.

P5, left column, 2nd paragraph, final sentence: not sure whether examination of flanking regions can truly help differentiate between decay rather than excision. What if excising TEs have a tendency to insert within TEs?

This sentence has been deleted.

P5, left column, 3rd paragraph, final sentence: correct typo 'from perfectly' to 'from being perfectly'

Both work, no change needed.

P7, left column, 4th paragraph: Please mention how methylation data is analyzed (BSseq reads remapped on each genome assembly, if I understood correctly)

Yes, the BSseq reads from Kawakatsu et al 2016 were remapped onto the corresponding genome assembly for each available accession. To save space, the mentioned section has largely become supplementary (see the "Silencing of sSVs" subsection in the "Mobilome" section of the Supplementary Note). We added a clearer description of data analysis to the main text and the Supplementary note, and added a reference to the Methods where this is explained.

P7, left column, 4th paragraph: 'sSVs containing TEs or TE fragments are more variable' in what sense? In terms of DNA methylation levels?

Sentence has been clarified.

P7, left column, 4th paragraph: 'consistent with a subset of these sSVs corresponding to un- or mis-annotated TEs' which subset? The ones that are highly methylated?

Yes, sentence has been clarified.

P7, left column, 5th paragraph: the claim that 'sSVs without annotated TE content behave similarly to sSVs containing TEs or TE fragments' is not at all what one extract from Extended Data Fig. 23. However there is indeed a subset of those that appear to behave similarly in terms of being highly methylated. Please correct accordingly.

Yes, that is what we meant: the whole section has been re-written.

P7, right column, 1st paragraph: Idem, with the claim to un-annotated sSVs that are part of the graph of nestedness are 'almost as highly methylated as previously annotated TEs'.

No, this is actually correct, but the claim has been moderated.

P7, right column, last paragraph: in the first version of this manuscript there was a hotspot in the lyrata genome of PC_new genes in 6C but they are gone now. Why? What did they correspond to? What were the changes responsible for their disappearance?

This was due to a single gene that upon further filtering was removed from the list of "PC_new genes".

Reviewer #3 (Remarks to the Author)

Thanks for providing a clear and precise answer to my previous comments, in a tone that sounds very human and not AI based. Very much appreciated.

Although much is yet to be described in terms of SVs, the paper represents a landmark.

I could not systematically check all the fixed bugs in the text but I trust the authors.

Congratulations to the authors for a great job. Looking forward to all follow-up papers.

Thank you!